# A metabolic synthetic lethality of phosphoinositide 3-kinase-driven cancer

Guillaume P. Andrieu [1,2] ✉, Mathieu Simonin [1,2,3], Aurélie Cabannes-Hamy[4], Etienne Lengliné[5], Ambroise Marçais [1,6], Alexandre Théron [7], Grégoire Huré[1,2], Jérome Doss[1,2], Ivan Nemazanyy [8], Marie Émilie Dourthe[1,2,9], Nicolas Boissel [5], Hervé Dombret[5], Philippe Rousselot[4], Olivier Hermine[6,10] & Vahid Asnafi [1,2] ✉

The deregulated activation of the phosphoinositide 3-kinase (PI3K) pathway is a hallmark of aggressive tumors with metabolic plasticity, eliciting their adaptation to the microenvironment and resistance to chemotherapy. A significant gap lies between the biological features of PI3K-driven tumors and the specific targeting of their vulnerabilities. Here, we explore the metabolic liabilities of PI3K-altered T-cell acute lymphoblastic leukemia (T-ALL), an aggressive hematological cancer with dismal outcomes. We report a metabolic crosstalk linking glutaminolysis and glycolysis driven by PI3K signaling alterations. Pharmaceutical inhibition of mTOR reveals the singular plasticity of PI3K-altered cells toward the mobilization of glutamine as a salvage pathway to ensure their survival. Subsequently, the combination of glutamine degradation and mTOR inhibition demonstrates robust cytotoxicity in PI3K-driven solid and hematological tumors in pre-clinical and clinical settings. We propose a novel therapeutic strategy to circumvent metabolic adaptation and efficiently target PI3K-driven cancer.

Aggressive cancers are a leading global cause of mortality and represent a major clinical challenge. Rapidly propagating tumors with highly plastic liabilities can adapt to severe environments and evade chemotherapies, conveying dismal outcomes for the patients. Among the common oncogenetic traits that favor outgrowth and refractoriness, the PI3K/Akt/mTOR pathway has significant implications in a wide range of solid and hematological cancers[1,2]. Activating alterations of the PI3K signaling are observed in the vast majority of human tumors, with central contributions to the regulation of cancer cell growth,

survival, metabolic adaptation, and chemoresistance[3]. The fundamental oncogenic roles of PI3K signaling have spurred considerable efforts to develop targeted therapies against the PI3K, Akt, and mTOR. While several compounds have shown promising efficacies in preclinical models, the clinical trials have dampened this enthusiasm, due to limited single-agent activity and accentuated toxicity in patients[2,4]. Critically, there is an unmet clinical need for several cancers with frequent PI3K signaling alterations with limited therapeutic arsenal, notably in the eventuality of chemoresistance and relapse. Amid these

[1]Laboratory of Onco-Hematology, Assistance Publique-Hôpitaux de Paris (AP-HP), Hôpital Universitaire Necker Enfants-Malades, Université Paris Cité, Paris, France. [2]Institut Necker-Enfants Malades (INEM), INSERM U1151 CNRS UMR8253, Université Paris Cité, Paris, France. [3]Department of Pediatric Hematology and Oncology, AP-HP, Hôpital Armand Trousseau, Université Paris Sorbonne, Paris, France. [4]Service d'Hématologie et d'Oncologie, Hôpital Universitaire de Versailles, APHP, Versailles, France. [5]Laboratory of Hematology and Institut de Recherche Saint-Louis EA3518, Hôpital Universitaire Saint-Louis, Université Paris Cité, Paris, France. [6]Service d'Hématologie Adulte, Hôpital Universitaire Necker-Enfants Malades, APHP, Université Paris Cité, Paris, France. [7]Department of Pediatric Oncology and Hematology, Hôpital Universitaire de Montpellier, Université de Montpellier, Montpellier, France. [8]Platform for Metabolic Analyses, Structure Fédérative de Recherche Necker, INSERM US24, CNRS UAR3633, Université Paris Cité, Paris, France. [9]Department of Pediatric Hematology and Immunology, AP-HP, Hôpital Universitaire Robert Debré, Université Paris Cité, Paris, France. [10]Department of Hematology, INSERM U1163, IMAGINE Institute, Hôpital Universitaire Necker Enfants-Malades, Université Paris Cité, Paris, France. ✉e-mail: guillaume.andrieu@inserm.fr; vahid.asnafi@aphp.fr

malignancies, T-cell acute lymphoblastic leukemia (T-ALL) constitutes a class of hematological cancers in children, adolescents and young adults, marked by aggressive behavior and poor clinical response, notably for relapsing cases despite salvage chemotherapy[5–9]. The oncogenic activation of PI3K signaling is observed in 15–20% of the patients and conveys a worse prognosis and dismal outcomes[10–15]. Yet, a significant gap subsists between the comprehension of PI3K-driven cancer biologic liabilities and the translation into efficient targeted therapies.

First coined by Otto Warburg, the avid reliance of tumor cells on glucose to suffice their energetical needs and provide substrates for biomass anabolism constitutes a hallmark of cancer[16,17]. Major advances have described the metabolic liabilities of tumor cells and the impact of their oncogenic lesions on their capacity to rewire metabolic networks[17–20]. Metabolic plasticity confers to cancer cells the ability to adapt to stressful environments as encountered during rapid outgrowth or under the pressure of therapy[21,22]. Major oncogenic drivers such as PI3K/Akt/mTOR pathway, Notch1, or c-Myc, contribute to metabolic rewiring and treatment failure[22–26]. A better understanding of the metabolic circuitries permitting such flexibility would pave the way to identifying new targets to hinder the metabolic adaptation of cancer cells and leverage the therapeutic response.

With the model of aggressive T-ALL, we identified a metabolic synthetic lethality revealed by PI3K addiction, that is conserved in a wide range of solid tumors. We propose a clinical-grade anti-metabolic strategy leading to an efficient targeted therapy in PI3K-driven cancers.

## Results

### PI3K signaling alterations define an aggressive subgroup of leukemia

Genetic alterations of PI3K signaling core genes *PTEN*, *PIK3R1*, *PIKCA*, and *AKT1* are frequent in both pediatric and adult T-ALL. We report their incidence and clinical impact in a series of 476 patients enrolled in the FRALLE2000 (*n* = 261) and the GRALL03-05 (*n* = 215) studies. Their baseline characteristics are summarized in Supplementary Table 1. The overall incidence of PI3K signaling alterations was 18.3% (89/476 patients) with similar frequencies in the pediatric and the adult cohorts (Fig. 1A and Supplementary Fig. 1A). *PTEN* loss of function accounted for the majority of these alterations and was detected in 14.6% (69/476) of the patients (Fig. 1B and Supplementary Fig. 2). Gain-of-function mutations of *PIK3R1*, *PIKCA*, and *AKT1* were sparsely detected (3.5%, 1.4%, and 0.4% respectively), and often co-occurred with *PTEN* inactivation (Fig. 1A). As previously described in T-ALL, PI3K signaling-altered patients had a poorer response to chemotherapy (Supplementary Table 1), a shorter overall survival compared to wild-type cases (Fig. 1B and Supplementary Fig. 1B), and higher incidences of disease relapse (Supplementary Fig. 1C).

The aggressiveness of PI3K-altered T-ALL is deeply imprinted in the blasts, as patient-derived xenografts (PDX) generated from fresh primary samples recapitulated their clinical characteristics marked by higher engraftment rates (Fig. 1C), and a rapid expansion in mice compared to wild-type cases (Fig. 1D, E). Remarkably, PI3K signaling

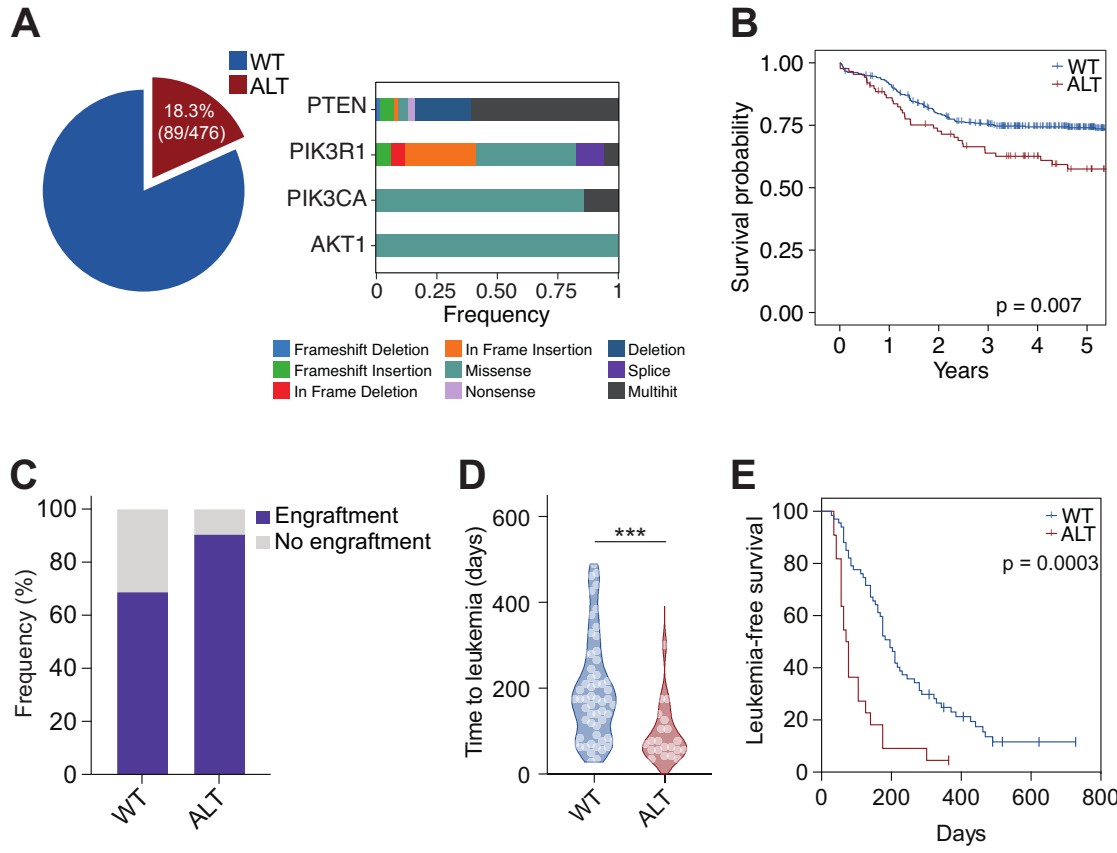

**Fig. 1 | PI3K signaling alterations define an aggressive subgroup of leukemia.**
**A** Incidence of PI3K signaling alterations in the GRAALL03-05 and FRALLE2000 cohorts. The nature of the oncogenetic lesions is indicated. **B** Overall survival of the patients stratified by their status regarding PI3K signaling (wild-type (WT) or altered (ALT)). Outcome comparisons were performed using a Cox regression analysis.
**C** Engraftment rates of primografts following the injection of fresh primary T-ALL samples into NSG mice evaluated from 101 primograft attempts. Successful engraftment rates are indicated (wild-type: 55/80 (68.8%) vs altered: 19/21 (90.5%) patients). **D** Time-to-leukemia analysis evaluated in PDX models as the time from injection to a leukemic burden >90% (wild-type (WT), *n* = 57 individual PDX; PI3K-altered (ALT), *n* = 21 individual PDX). An unpaired Mann-Whitney test was run (***, *p* < 0.001). **E** Leukemia-free survival of the PDX using the same criterion. Outcome comparisons were performed using a Cox regression analysis.

alterations were stably conserved from primary samples to PDX, indicating that these oncogenic lesions drive T-ALL progression (Supplementary Fig. 2). Altogether, PI3K-altered T-ALL defines an aggressive subgroup with inferior outcomes.

### Aberrant PI3K signaling polarizes leukemia metabolism towards glycolysis

The oncogenic activation of the PI3K signaling sustains leukemia expansion via multiple mechanisms including cell growth, proliferation, metabolism, and chemosensitivity modulation. We screened a series of PDX by phosflow to determine PI3K signaling activity. Sustained phosphorylation of Akt (pAkt$^{S473}$) and subsequent mTOR target S6 (pS6$^{T235/236}$) were detected in PI3K-altered PDX (Fig. 2A). The mTOR target 4E-BP1 (p4E-BP1$^{T37/46}$) was not found differentially activated between the two groups. These data confirm the robust oncogenic activation of PI3K and downstream master regulator mTOR in this subgroup in comparison with wild-type cases.

To further explore the transcriptomic and metabolic profile of PI3K-altered T-ALL, we analyzed a series of 155 primary T-ALL samples by RNA sequencing. Differential expression analysis performed on a subset of 2,141 expressed metabolic genes revealed an upregulation of transcripts related to glycolysis and TCA cycle in PI3K-altered T-ALL (Fig. 2B). Conversely, genes encoding for enzymes and regulators of nicotinate, amino acids, and steroids were found downregulated. Enrichment analyses carried out on differentially expressed metabolic genes in PI3K-altered T-ALL versus wild-type cases revealed that upregulated genes were mapped to the TCA cycle, the oxidative phosphorylation and the electron transport chain, and the fatty acid metabolism, three major metabolic pathways strongly wired to glycolysis and its derivates (Fig. 2C), along with mTORC1 signaling. Downregulated genes were related to steroids and cholesterol synthesis and amino acid metabolism. Gene set enrichment analysis confirmed that oxidative phosphorylation and fatty acid metabolism are two important features of PI3K signaling-altered T-ALL (Fig. 2D).

Given the preponderant role of PI3K/Akt/mTOR in the control of glucose metabolism[3,10,23], we evaluated the glycolytic activity in our settings. Glucose uptake was found to increase in PI3K-altered T-ALL cell line models *versus* wild-type, indicative of enhanced glycolytic activity (Fig. 3A). Hence, we examined the sensitivity of PI3K-altered T-ALL to glucose deprivation. While wild-type T-ALL cell lines tolerated glucose limitation, PI3K-altered cell lines did not survive (Fig. 3B), highlighting the strong reliance on glucose to sustain the energetical needs of these models. Importantly, PDX models carrying PI3K signaling alterations also presented a stronger glucose consumption than wild-type cases (Fig. 3C). Robust correlations were observed between the phosphorylation of PI3K signaling effectors and glucose uptake in both wild-type and PI3K-altered PDX (pAkt: $R^2 = 0.659$, $p = 0.0004$ in WT, $R^2 = 0.922$, $p = 0.0001$ in ALT; pS6: $R^2 = 0.501$, $p = 0.0046$ in WT, $R^2 = 0.533$, $p = 0.0039$ in ALT) (Fig. 3D).

To further validate that PI3K-altered blasts actively consume glucose, metabolic tracing analysis using uniformly labeled glucose was performed ex vivo on PDX blasts after a 16 h glucose deprivation. Time-course evaluation of the incorporation of 13C-glucose into glycolysis intermediates confirmed that PI3K-altered blasts actively rely on glycolysis (Supplementary Fig. 3). This incorporation was found more rapid than in wild-type blasts. These results reflect the coupling of PI3K signaling and glucose consumption in T-ALL. Hence, we tested whether PI3K targeting could hinder glucose accessibility to leukemic blasts. Both PI3K and mTORC1 inhibition by wortmannin and temsirolimus respectively reduced glucose consumption in PI3K-driven blasts, confirming the modulation of glycolysis by sustained PI3K/Akt/mTOR signaling (Fig. 3E). However, despite elevated signaling activity, pharmaceutical inhibition of PI3K or mTORC1 demonstrated poor

efficacy ex vivo on PI3K-altered blasts (Fig. 3F), reinforcing the concept that the sole inhibition of PI3K signaling components has limited efficacy in cancer[2,4].

We thus examined the capacity of PI3K-altered blasts to survive glucose limitation. In sharp contrast with T-ALL cell lines models, glucose deprivation of PDX only had limited cytotoxicity and moderate reduction of blast proliferation with PI3K signaling alterations (Fig. 3G, H). Furthermore, glucose limitation did not impair the strong signaling activation seen in PI3K-altered blasts, as phosphorylation levels of Akt and S6 remained elevated under glucose deprivation (Fig. 3I). Likewise, ATP levels in these blasts remained unchanged under glucose deprivation (Fig. 3J), suggesting that despite elevated basal glycolytic rates, blasts with aberrant activation of the PI3K pathway can use alternative metabolic pathways to cope with glucose limitation and maintain their energy balance. Hence, the metabolic plasticity driven by PI3K signaling alterations may critically contribute to the survival of primary leukemic blasts in nutrient-deprived environments.

### The metabolic adaptation of PI3K-altered leukemia relies on glutaminolysis

To explore the metabolic circuitries of PI3K-altered T-ALL sustaining blast survival upon glucose limitation, we conducted a metabolome profiling of 14 PDX (7 WT, 7 ALT) cultured ex vivo in complete medium or without glucose. As expected, glucose deprivation drastically suppressed glycolysis intermediates, irrespective of the mutational status of PI3K signaling (Fig. 4A). Of most importance, glutamine and its derivate glutamate were found consumed in PI3K-altered T-ALL, while remaining unchanged in wild-type cases. These results strongly suggest that glutamine mobilization participates in the survival of PI3K-altered blasts when glucose availability is limited. Enrichment analyses on the pool of deregulated metabolites upon glucose depletion confirmed the downregulation of glycolysis. Related metabolic pathways such as the pentose phosphate pathway, pyruvate metabolism, and TCA cycle were also found suppressed in wild-type samples (Fig. 4B). Importantly, glutaminolysis was confirmed as a hallmark of PI3K-altered T-ALL upon glucose limitation. While glycolysis intermediates and products pyruvate and lactate were drastically reduced under glucose deprivation in both groups, glutamine, and glutamate were only found consumed in PI3K-altered blasts (Fig. 4C). Of note, while TCA cycle intermediates such as α-ketoglutarate, fumarate, and citrate were found reduced in wild-type samples, these metabolites were unchanged in PI3K-altered blasts, confirming the mobilization of glutamine in an anaplerotic flux to support the TCA in these conditions (Fig. 4C). To validate the metabolic plasticity between glycolysis and glutaminolysis in PI3K-altered T-ALL, we first measured glucose uptake in blasts either starved for glucose or glutamine. Wild-type blasts did not significantly change their glucose consumption in deprived conditions. However, glucose uptake was markedly increased as a response to glutamine limitation (Supplementary Fig. 4A). Second, metabolic tracing was performed to evaluate the contribution of glutamine to sustain the TCA upon glucose limitation. Time-course tracing revealed active glutaminolysis upon glucose deprivation in PI3K-altered blasts with the incorporation of labeled carbons from glutamine to glutamate, a-ketoglutarate, and TCA intermediates (Fig. 4D). In contrast, limited glutaminolytic activity was measured in PI3K-WT blasts. Of note, alternate glutamine metabolisms such as reductive carboxylation or de novo asparagine synthesis were not found induced in PI3K-altered blasts (Supplementary Fig. 4B), reinforcing glutaminolysis as the main metabolic route balancing glucose deprivation in these cells. Altogether, our results identified glutamine and glutaminolysis as a metabolic circuitry activated in PI3K-altered blasts upon glucose deprivation to support their survival.

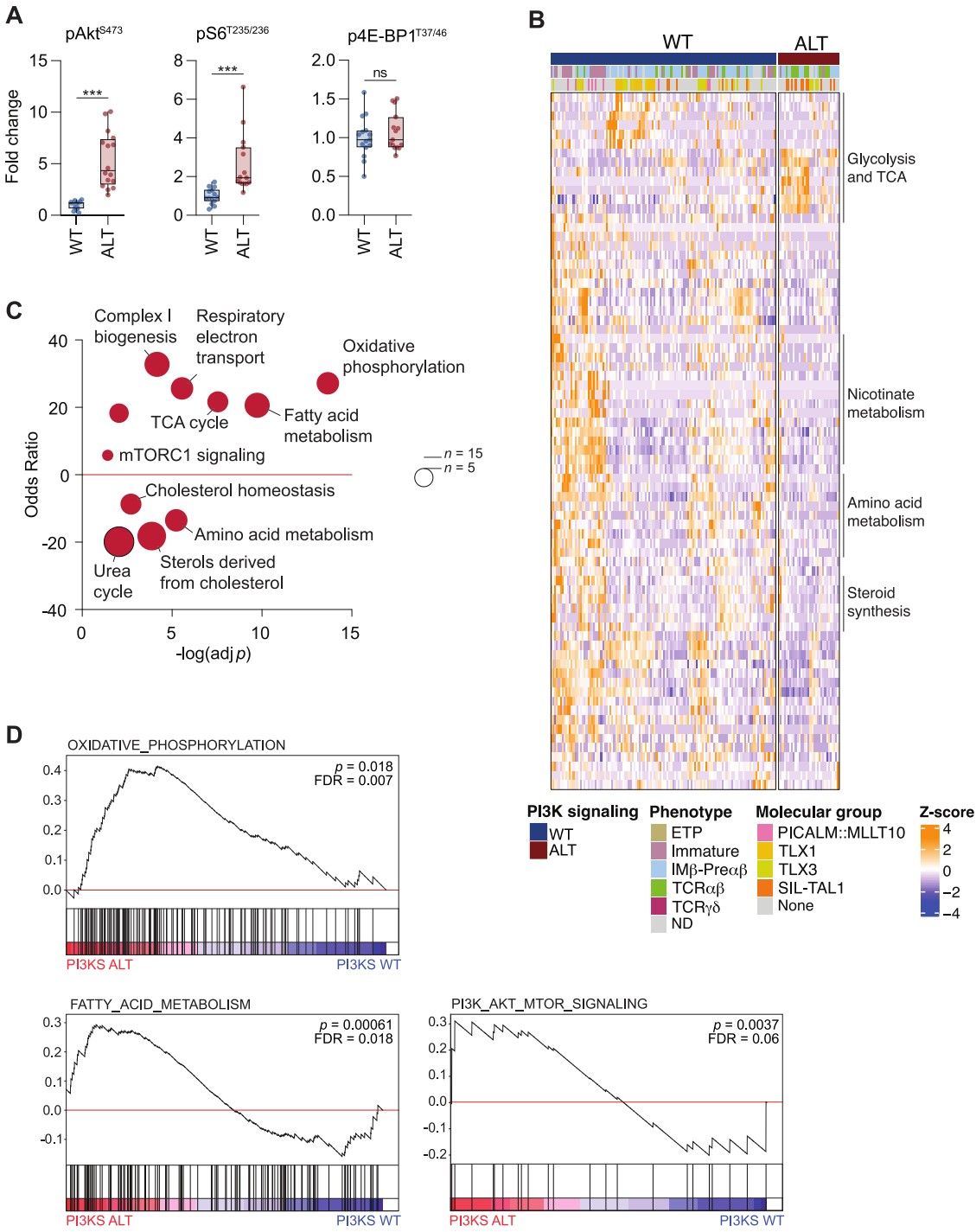

**Fig. 2 | Sustained PI3K signaling activation is coupled with glycolytic and mitochondrial metabolism transcriptomic signatures. A** Phosflow evaluation of PI3K signaling activity in leukemic blasts (wild-type: $n = 16$ PDX, PI3K-altered: $n = 16$ PDX). Box plots show minima, maxima, median, and 25% and 75% percentiles. Unpaired Welch-corrected two-tailed t-test (pAkt$^{S473}$) and two-tailed Mann-Whitney tests (pS6$^{S235/236}$, p4E-BP1$^{T37/46}$) were run (ns, $p > 0.05$; ***, $p < 0.001$). **B** Heatmap of expression by transcriptomics and clustering analysis of the 75 metabolic genes differentially expressed in PI3K-altered vs wild-type patients (WT: 122 patients, ALT: 33 patients). Differential expression was defined by an absolute log2 fold change >1 and an adjusted $p$ value < 0.05 from DESeq2 analysis (Wald test-adjusted for multiple comparisons). The metabolic pathways enriched in the gene clusters are listed. **C** Pathway enrichment analysis in PI3K-altered vs wild-type T-ALL samples using the 75 significantly deregulated metabolic genes in PI3K-altered vs WT samples. An over-representative analysis adjusted for multiple comparisons was run with MetaboAnalyst. A positive odds ratio indicates an enrichment in in PI3K-altered samples. Bubble sizes indicate the number of deregulated genes mapped to each pathway. **D** Gene set enrichment analyses based on the expression of 2141 metabolic genes in 155 patients stratified by their PI3K signaling status (GSEA-based over-representative analysis adjusted for multiple comparisons).

## Glucose limitation reveals a synthetic lethality to glutamine targeting in PI3K-driven leukemia

We sought to functionally demonstrate the importance of glutamine mobilization to rescue glucose limitation in PI3K-altered T-ALL. The conversion of glutamine into glutamate by glutaminases (GLS) is the first and rate-limiting step of glutaminolysis. Hence, we evaluated the sensitivity to GLS inhibitor CB-839. While moderate cytotoxicity was observed ex vivo in complete medium conditions, GLS inhibition

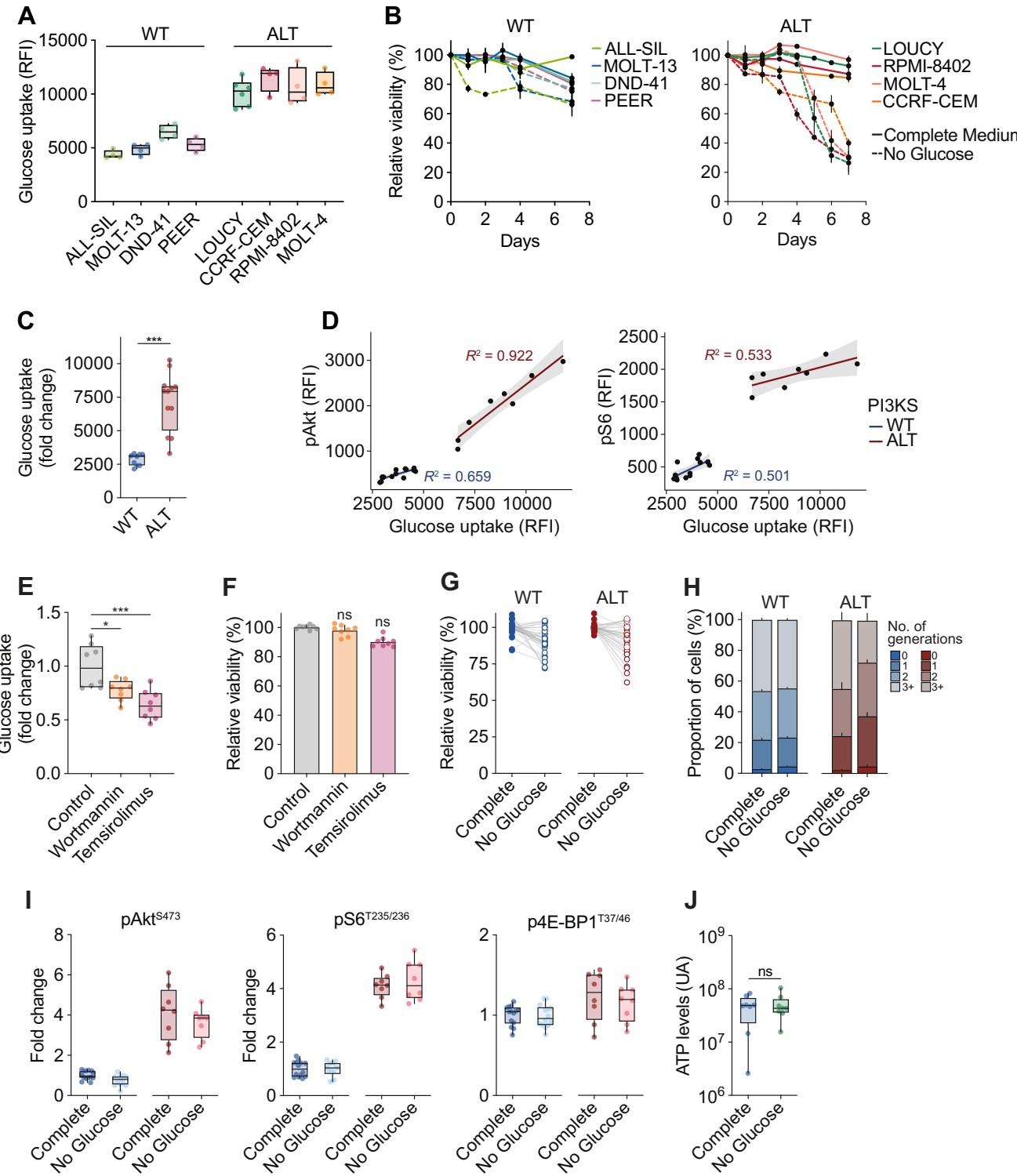

efficiently eradicates blasts in the absence of glucose in PI3K-altered blasts (Fig. 5A), supporting that glutamine mobilization sustains the adaptation of blasts to glucose deprivation. Since the glycolytic flux is controlled by mTORC1 (Fig. 3D, E), we examined whether mTOR inhibition, by mimicking the effect of glucose limitation, synergizes with glutaminolysis inhibition in PI3K-altered T-ALL. As previously observed, mTORC1 inhibition by temsirolimus showed limited cyto-toxicity (Fig. 3G and Fig. 5B). Of note, similar results were obtained with various inhibitors of mTORC1 or/and mTORC2 (data not shown), indicating that PI3K-altered blasts can resist mTOR inhibition. In sharp contrast, temsirolimus had a strong cytotoxic effect upon glutamine

limitation (Fig. 5B). Additionally, while single inhibition of either glu-taminolysis or mTOR was inefficient, the combinatory treatment showed a strong synergy specifically in PI3K-altered blasts (Fig. 5C, D). Hence, our data illustrate the metabolic crosstalk between glycolysis and glutaminolysis in PI3K-driven T-ALL.

L-asparaginase is commonly used in the treatment of T-ALL exploiting the inability of the blasts to synthesize de novo asparagine due to the lack of asparagine synthetase (ASNS)[27,28]. *Erwinia chry-santhemi*-derived L-asparaginase erwinase has a strong glutaminase activity and can deplete both extracellular asparagine and glutamine[29]. While wild-type PDX displayed a highly heterogeneous response to

**Fig. 3 | Aberrant PI3K signaling polarizes leukemia metabolism towards glycolysis. A** Glucose uptake in T-ALL cell lines. Each dot represents a replicate. Box plots show minima, maxima, median, and 25% and 75% percentiles. **B** Cell viability at 72 h of wild-type and PI3K-altered T-ALL cell lines under complete culture conditions or without glucose (mean ± SEM are indicated per cell line from three replicates in three independent experiments). **C** Glucose uptake in wild-type (*n* = 9 individual PDX) and PI3K-altered (*n* = 12 individual PDX) T-ALL PDX. Box plots show minima, maxima, median, and 25% and 75% percentiles. An unpaired Welch-corrected two-tailed t-test was run (***, *p* < 0.001). **D** Pearson correlations between phosflow p-Akt or p-S6 and glucose uptake levels in wild-type (*n* = 14 individual PDX) PI3K-altered (*n* = 8 individual PDX) T-ALL PDX. The SEM is indicated by the gray area. **E** Glucose uptake in PI3K-altered PDX was measured at 72 h upon PI3K signaling inhibition (*n* = 8 individual PDX). Box plots show minima, maxima, median, and 25% and 75% percentiles. A one-way ANOVA test adjusted for multiple comparisons (Dunn–Šidák correction) was run (*, *p* < 0.05; ***, *p* < 0.001). **F** Cell

viability at 72 h of PI3K-altered PDX was measured upon PI3K signaling (mean ± SEM, *n* = 8 individual PDX). A one-way ANOVA adjusted for multiple comparisons (Dunn's test) was performed (ns, *p* > 0.05). **G** Cell viability at 72 h of wild-type and PI3K-altered T-ALL PDX under complete culture conditions or without glucose (*n* = 18 individual PDX evaluated in duplicates). **H** Cell proliferation of CellTrace-labelled wild-type and PI3K-altered T-ALL PDX was evaluated after 72 h in complete culture condition or without glucose (mean ± SEM, *n* = 4 individual PDX). Each generation was defined by a CellTrace peak. **I** Phosflow evaluation of PI3K signaling activity in wild-type (*n* = 14) and PI3K-altered (*n* = 8) blasts cultured for 72 h in complete medium or without glucose. Box plots show minima, maxima, median, and 25% and 75% percentiles. **J** ATP levels in PI3K-altered PDX were evaluated after 72 h in complete culture condition or without glucose (*n* = 8 individual PDX) by LC/MS. Relative abundances are indicated. Box plots show minima, maxima, median, and 25% and 75% percentiles. An unpaired homoscedastic two-tailed t-test was performed (ns, *p* > 0.05).

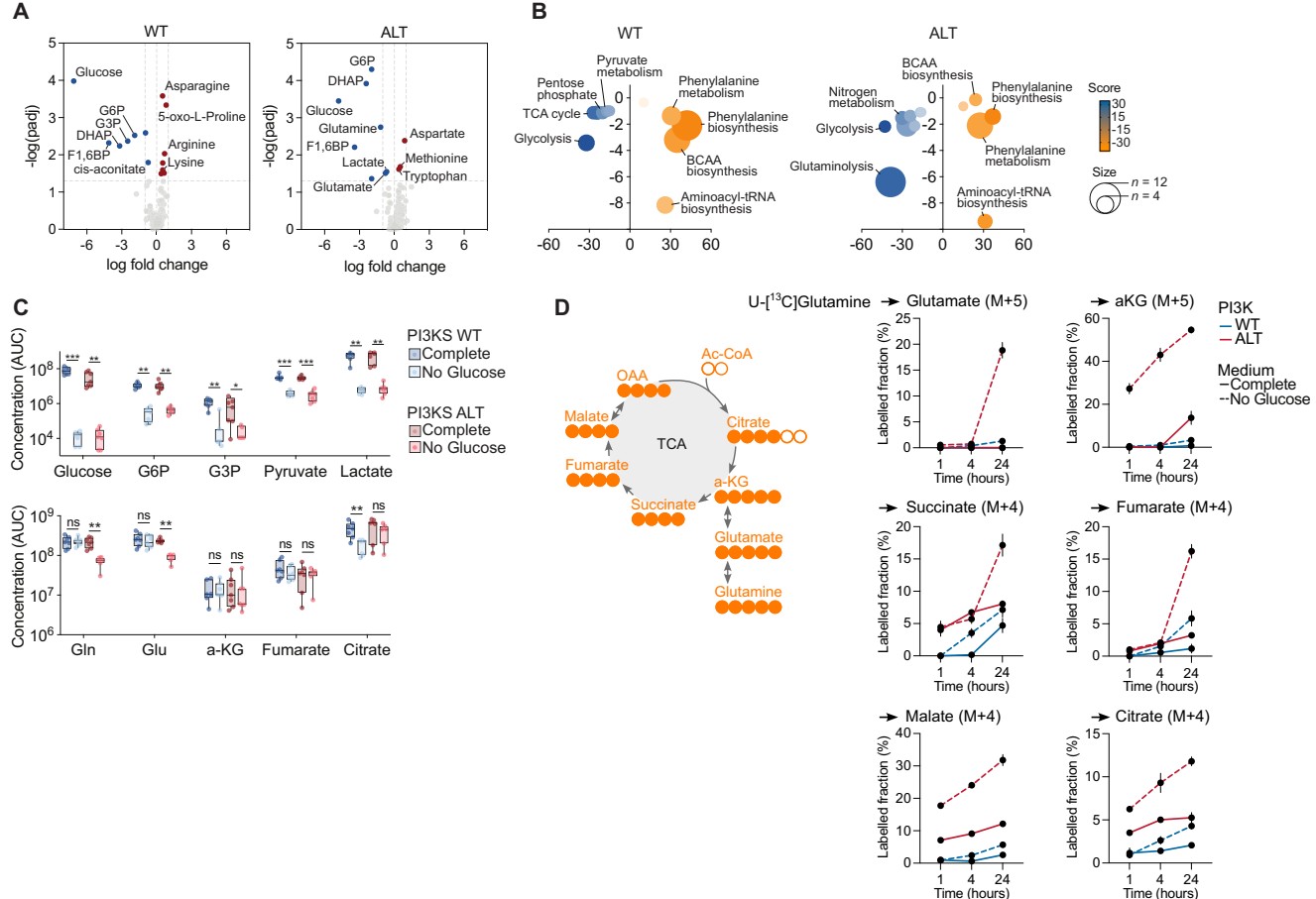

**Fig. 4 | PI3K-altered leukemias rely on glutamine metabolism to cope with glucose limitation. A** Volcano plots indicating differential metabolite fold change in wild-type (*n* = 7 individual PDX) and PI3K-altered (*n* = 7 individual PDX) blasts cultured for 72 h without glucose vs complete medium. For each genotype, a two-tailed unpaired homoscedastic t-test was performed. Adjusted *p* values were used. **B** Bubble plots depicting metabolic pathway modulation upon glucose deprivation. The size of the bubbles is proportional to the number of differentially detected metabolites mapped to each pathway (x-axis: score, y-axis: -log₁₀(pajd)). An over-representative analysis adjusted for multiple comparisons was run with MetaboAnalyst. **C** Relative metabolite concentrations in wild-type (*n* = 7 individual PDX) and PI3K-altered (*n* = 7 individual PDX) PDX blasts cultured for 72 h without glucose vs

complete medium. Relative abundances are indicated. Box plots show minima, maxima, median, and 25% and 75% percentiles. Two-way ANOVA tests adjusted for multiple comparisons (Tukey correction) were run (ns, *p* > 0.05; *, *p* < 0.05; **, *p* < 0.01; ***, *p* < 0.001). **D** Metabolic tracing with ¹³C5-glutamine performed on PI3K wild-type (*n* = 3 individual PDX) and PI3K-altered (*n* = 3 individual PDX) T-ALL PDX blasts cultured for 24 h in complete or glucose-free media then incubated with 2 mM ¹³C5-glutamine for the indicated times. Left. Schematic representation of the labeling strategy. Right. Labeled fraction of ¹³C5-glutamine-derived metabolite. Each analyzed isotopologue is indicated. Each dot represents the average ± SEM of three biological replicates of the three samples per condition.

erwinase alone or combined with temsirolimus, a strong cytotoxic effect was observed for the combination in all PI3K-altered blasts (Fig. 5E). Importantly, the use of another L-asparaginase devoid of glutaminase activity kidrolase failed to reproduce these results and

poorly synergizes with temsirolimus (Fig. 5F). Indeed, while both L-asparaginases efficiently depleted extracellular and intracellular asparagine to produce L-aspartic acid, only erwinase additionally reduces glutamine pools and increases the availability of glutamate

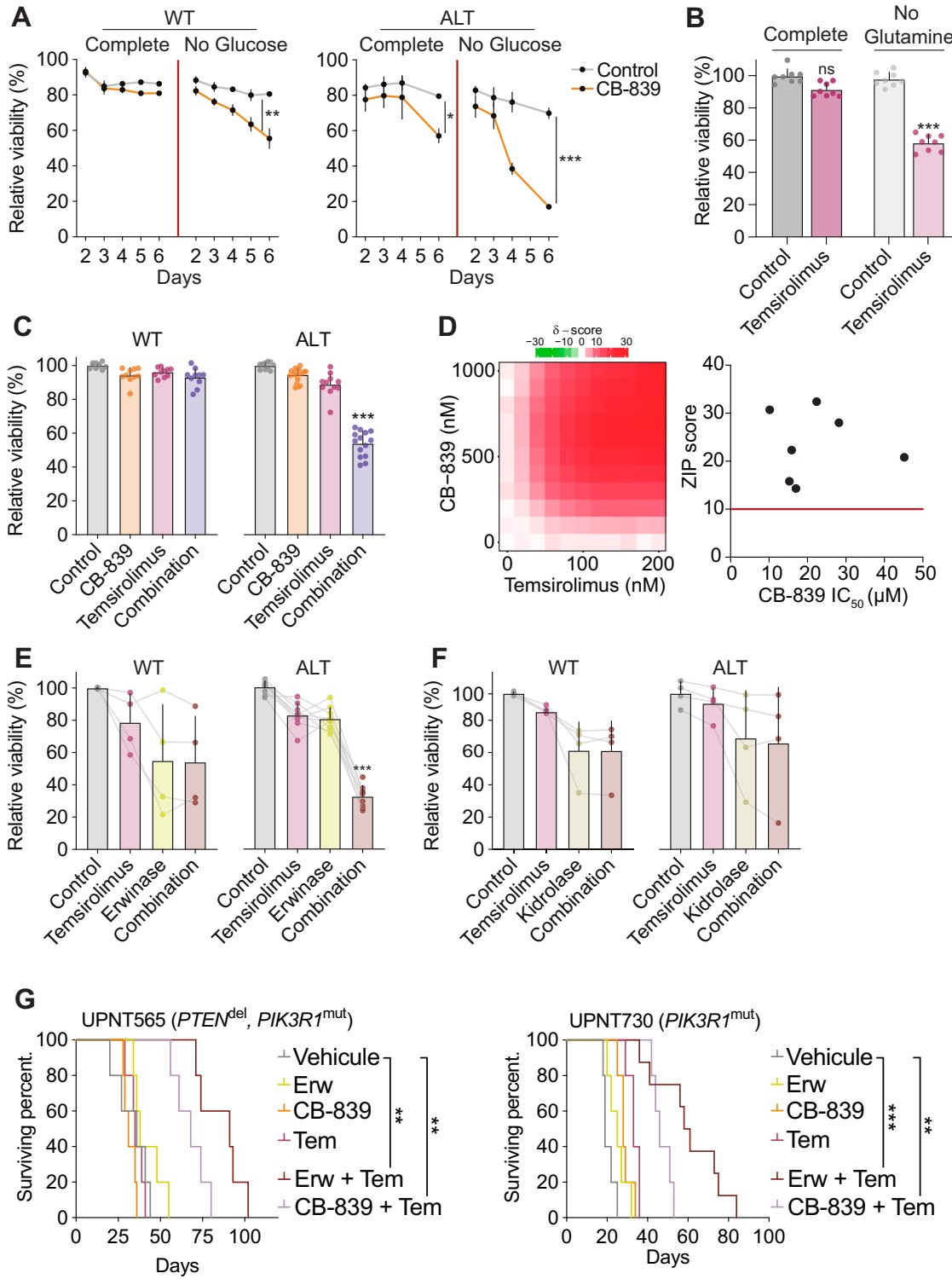

(Fig. 5G). These data strongly suggest that the efficacy of erwinase combined with temsirolimus strongly relies on its glutaminolytic activity. Taken together, these results strongly support the combination of mTOR inhibitors and glutamine metabolism targeting compounds such as CB-839 or erwinase in the treatment of PI3K-altered T-ALL. To address this, we tested this approach in vivo on PI3K-altered xenografts in a pre-clinical setting of two cycles of five-day-long daily gavage with temsirolimus and/or intravenous injection of erwinase every two days. The aggressive behavior of this subgroup of T-ALL was exemplified by the drastically limited survival seen in the vehicle arm (Fig. 5H). While nor erwinase neither temsirolimus treatment did not

reduce the tumor burden or prolong survival, the combined treatment resulted in a marked improvement in the control of tumor outgrowth and the survival of mice.

## Clinical evaluation of the erwinase-temsirolimus association in PI3K-driven leukemia

We report the clinical course and outcomes of five patients suffering from refractory or relapse (R/R) T-ALL or T-LL (T-ALL/LL) harboring PI3K-altered and treated with the association erwinase and temsirolimus (ET). Four of these patients were enrolled in the ALL TARGET OBS, a national registry focused on R/R T-ALL adult patients

**Fig. 5 | Glucose limitation reveals a synthetic lethality to glutamine targeting in PI3K-driven leukemia. A** Relative cell viability of wild-type PI3K ($n = 14$ individual PDX) and PI3K-altered ($n = 8$ individual PDX) PDX blasts after 72 h of culture in complete or glucose-free medium with control (DMSO) or CB-839 1 μM treatment (mean + SEM are indicated). One-way ANOVA tests adjusted for multiple comparisons (Šidák correction) were run (*, $p < 0.05$; **, $p < 0.01$; ***, $p < 0.001$). **B** Relative cell viability of PI3K-altered ($n = 8$ individual PDX) PDX blasts after 72 h of culture in complete or glutamine-free medium with control (DMSO) or temsirolimus 200 nM treatment (mean + SEM are indicated). A one-way ANOVA test adjusted for multiple comparisons (Dunn–Šidák correction) was run (ns, $p < 0.05$; ***, $p < 0.001$). **C** Relative cell viability of wild-type PI3K ($n = 14$ individual PDX) and PI3K-altered ($n = 8$ individual PDX) PDX blasts after 72 h of culture treated with control (DMSO), CB-839 1 μM, temsirolimus 200 nM or combined treatment (mean + SEM are indicated, each dot is a biological replicate). A one-way ANOVA test adjusted for multiple comparisons (Dunnett correction) was run (***, $p < 0.001$). **D** Left: Heatmap

depicting the Bliss synergy score of the combination CB-839 + temsirolimus computed from viability data of PI3K-altered PDX blasts ($n = 7$ individual PDX). Right: individual Bliss scores mapped to matched CB-839 IC50 values. A Bliss score >10 indicates synergy. **E** Relative cell viability of wild-type PI3K ($n = 4$ individual PDX) and PI3K-altered ($n = 8$ individual PDX) PDX blasts after 72 h of culture treated with control (DMSO), temsirolimus 200 nM, erwinase 1 U/ml or combined treatment (mean + SEM are indicated). A Kruskal-Wallis test adjusted for muliple comparisons (Dunn correction) was run (***, $p < 0.001$). **F** Relative cell viability of wild-type PI3K ($n = 4$ individual PDX) and PI3K-altered ($n = 4$ individual PDX) after 72 h of culture treated with control (DMSO), temsirolimus 200 nM, kidrolase 1 U/ml or combined treatment (mean + SEM are indicated). A Kruskal-Wallis test adjusted for multiple comparisons (Dunn correction) was run. **G** Survival curves of mice xenografted with two PI3K-altered PDX (5 mice/arm/PDX) and treated with vehicule, erwinase, CB-839, temsirolimus, or the indicated combinations. Outcome comparisons were performed using a Cox regression analysis (**, $p < 0.01$; ***, $p < 0.001$).

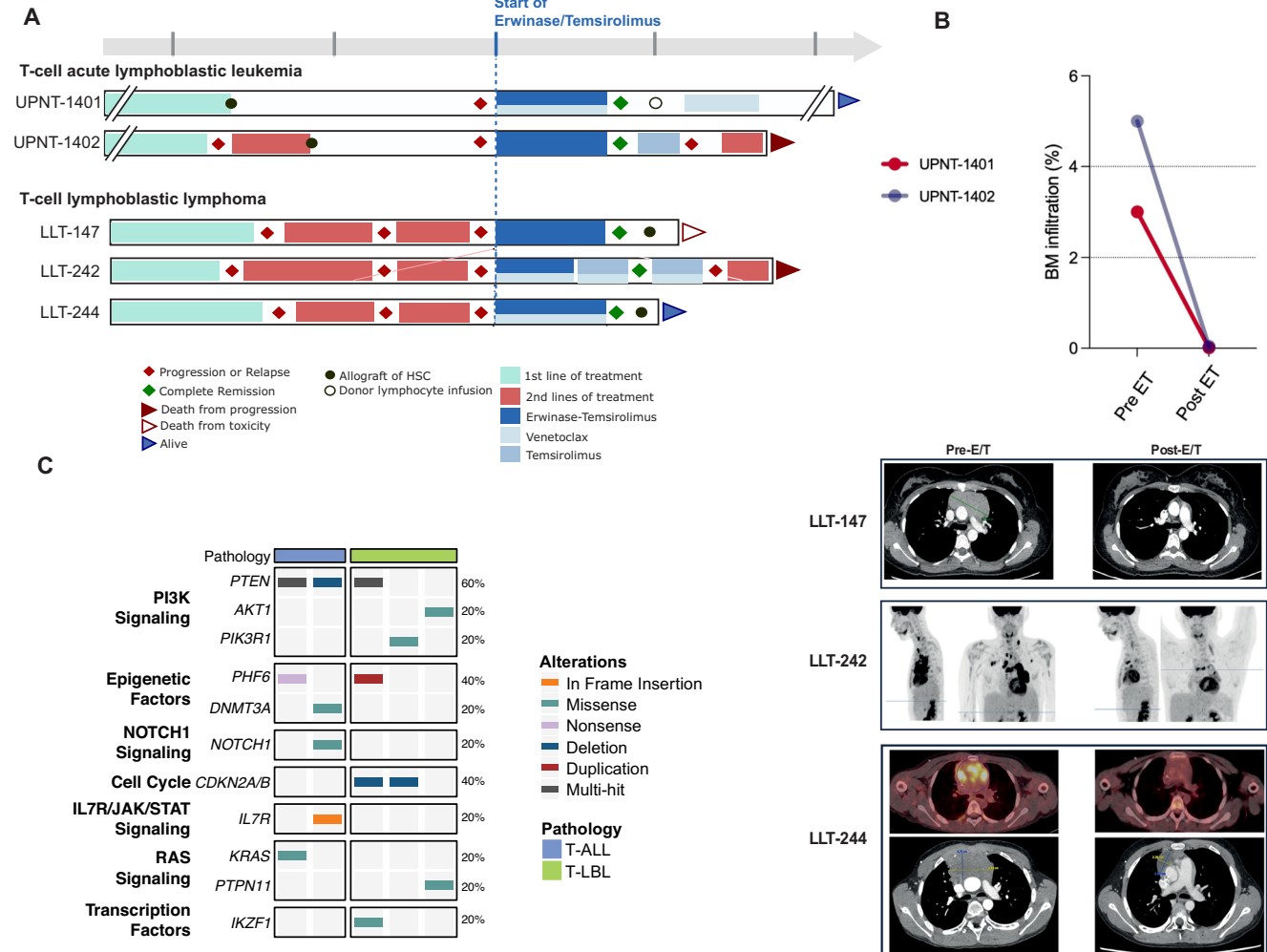

**Fig. 6 | Clinical evaluation of the erwinase-temsirolimus combination in PI3K-driven leukemia. A** Swimmer plots presenting the clinical history, treatment course, response, and outcome of the patients. **B** Bone marrow response, chest CT, and FDG-ET/CT evaluation of two patients presented a significant reduction in the

tumor burden following treatment with the erwinase-temsirolimus association. Each dot represents a patient. **C** Oncoplot presenting the main oncogenetic characteristics of the treated patients.

(NCT05832125). The primary objective of this registry is to assess the potential benefits of targeted therapies compared to a standard of care and to enhance access to innovation and personalized medicine for these patients facing a very poor prognosis. The fifth pediatric case was treated upon the approval of a special commission for off-label use and is included in a specific database (IRB #DC-2015-2473).

All five patients had a complex medical history and received multiple lines of treatment with marked resistance at the time of ET initiation (Fig. 6A). The median age at ET initiation was 26 years (8-50 years) and the sex ratio (M/F) was 3/2. All patients harbored PI3K signaling alterations with additional oncogenetic lesions (Fig. 6B). Among the five patients, three suffered from T-LL and two from T-ALL. The two

patients with T-ALL suffered from relapses following allogeneic stem cell transplantation (ASCT), while the three remaining patients with T-LL received up to three different lines of polychemotherapy before ET treatment (Fig. 6 and Supplementary Table 2). Based on the limited but promising previously published results that reported the efficacy of venetoclax-based therapies in R/R T-ALL/LL, three patients also received venetoclax combined with the ET association. All five patients achieved a response within one month after one or two courses. The two patients suffering from relapsed T-ALL achieved negative minimal residual disease (MRD) and patients with R/R T-LL experienced a significant decrease of the mediastinal mass based on early volumetric and/or metabolic evaluation assessment (Fig. 6A). Four out of five patients experienced expected and acceptable side effects related to the ET combination. However, the fifth patient experienced significant and severe toxicities (hematological, digestive, and infectious adverse events) that led to the discontinuation of the erwinase injection (UPNT-1402). Overall, the ET (± venetoclax) association was well tolerated. After achieving CR, three patients received consolidation therapy consisting of allogeneic SCT in two patients (LLT-147, LLT-244) while the previously allografted third patient (UPNT-1401) received donor lymphocyte infusion (DLI). After achieving CR post-ET therapy, one patient who received ASCT eventually died from allograft-related toxicity (LLT-147, acute respiratory distress syndrome). The patient who received a DLI remains in CR four years after the infusion (UPNT-1401), and so does the third patient who was recently allografted (LLT-244). For the two remaining patients, erwinase was discontinued due to toxicity for UPNT-1402, and to supply problems for LLT-242. These two patients rapidly progressed after the discontinuation of ET treatment and ultimately died from disease progression.

It is noteworthy that three out of five patients received a combination of ET with venetoclax. Since venetoclax has demonstrated efficacy in T-ALL, we cannot exclude that a part of the observed therapeutic effect may be attributed to its inclusion in the regimen. Yet, the T-ALL patient (UPNT-1402) obtained a negative MRD, and the T-LL patient (LLT-147) experienced a 75% reduction of the mediastinal mass, while not receiving venetoclax, demonstrating the efficacy of the ET regimen without BH3 mimetic. Importantly, the patient (LBL-T 242) who initially responded to the regimen ET with venetoclax experienced disease progression after erwinase discontinuation (due to supply limitations), despite a continued treatment with temsirolimus and venetoclax. This pinpoints the crucial role of erwinase in the chemotherapy regimen. Further investigations into the potential additive effect of venetoclax when combined with erwinase and temsirolimus are needed. Taken together, we unveiled a promising treatment combining an asparagine and glutamine degrader with a PI3K signaling inhibitor that should be considered as a therapeutic option in a bridge-to-transplant approach for R/R T-ALL/LL harboring PI3K signaling deregulation.

### PI3K-driven cancers are vulnerable to the dual targeting of glucose and glutamine

As the alterations of PI3K signaling are frequently observed in cancer beyond hematological malignancies, we screened a large panel of cancer cell lines to explore whether this metabolic vulnerability could be a feature of PI3K-dysregulated cancers. Single-agent treatment with temsirolimus had a minor impact on the different cancer cell lines tested (Supplementary Fig. 3A). As in T-ALL, solid cancer cells can be sensitive to the neutralization of Asn by L-asparaginases (Supplementary Fig. 3A). To better describe the specific cytotoxicity of the combination, it was evaluated compared with the sensitivity to L-asparaginases alone. Strikingly, a net cytotoxic effect of ET was observed in 13/16 PI3K-altered cancer cell lines (Fig. 7A, B). A robust correlation was observed between the cytotoxicity of ET ($r = 0.8036$, $p = 0.0002$) and the synergy of erwinase and temsirolimus (Fig. 7A-D). The most responsive were prostate, T-ALL, breast, and colon cancer

cell lines, with moderate effects in lung, melanoma, and pancreatic cancer cells. In sharp contrast, a limited effect was observed on wild-type cell lines from the same panel of cancers. Additionally, on asparaginase-resistant cell lines, kidrolase failed to be as efficient as erwinase, reinforcing that the glutaminase activity of erwinase may be integral to the cytotoxicity of the combination with temsirolimus.

Altogether, our data suggest that the oncogenic dysregulation of PI3K signaling may confer a unique metabolic profile that is cancer-agnostic and may be exploited in other PI3K-driven tumors. Metabolic plasticity conveyed by oncogenic lesions constitutes a targetable vulnerability in PI3K-driven leukemia that show promising results in preclinical and clinical settings.

## Discussion

The major metabolic hallmarks of tumor cells are the Warburg effect and the dependency on glutamine[16,19,20]. Lactic fermentation as an endpoint of glycolysis ensures continuous ATP production for cancer cells independently of other variables like oxygen availability, but it comes with a cost. Conversion of glucose to lactate generates ATP and pools of metabolic intermediates required for nucleotide synthesis, reductive biosynthesis, anti-oxidative responses, DNA methylation, hence sustaining anabolic pathways, biomass production, and cell proliferation, but lactate secretion eliminates carbon sources from the cell. Pyruvate conversion into lactate limits acetyl-CoA production available to fuel the TCA cycle. To compensate for this, cancer cells often deaminate glutamine into glutamate and subsequent a-ketoglutarate to produce an anaplerotic flux that replenishes the TCA cycle. Hence, tumor cells have imprinted metabolic flexibility that connects glucose consumption and glutamine mobilization. While being a non-essential amino acid, glutamine is recognized as one of the most critical metabolites for cancer cells[18,30]. Outgrowing tumor cells engulf more glutamine than they produce, rendering them highly dependent on the available extracellular pools of this amino acid in their microenvironment. Many oncogenes that drive oncogenesis and progression rely on glutamine metabolism. For instance, aberrant Notch1 activation, coupled with downstream effector c-Myc, sustains glutamine uptake and glutaminolysis machinery[23,31,32]. Anti-Notch1 interventions such as gamma-secretase inhibitors can limit glutamine mobilization but have demonstrated limited efficacy and strong toxicities, undermining their clinical consideration. The mutational landscape of T-ALL is complex, resulting in the co-occurrence of several oncogenes activation and tumor suppressor gene silencing[33]. *PTEN* inactivation has been linked to promoting metabolic plasticity and confers exacerbated glycolytic capacities to leukemic blasts[23]. In another model of colorectal cancer, the oncogenic activation of *PIK3CA* has been shown to reprogram cell metabolism to promote glutamine usage[24]. *PIK3CA* encodes the catalytic subunit of the isoform PI3Kα. While rarely found altered in T-ALL (1.5% of mutated cases in our cohort), we have recently reported that *PIK3CA* mutations are more frequent in T-LL, a hematological malignancy closely related to T-ALL[34]. In line with this, our results unravel metabolic plasticity conveyed by alterations of PI3K signaling between glucose and glutamine usage when one metabolite becomes limiting. Interestingly, it appeared that wild-type cancer cells were less polarized towards this singular metabolic interplay. Notably, the autophagic response upon glucose limitation may be an important PI3K-driven vector of metabolic adaptation towards glutamine.

Based on this metabolic singularity, we proposed a novel treatment combining an asparagine and glutamine degrader with a PI3K signaling inhibitor that presents promising efficacy in T-ALL/LL harboring PI3K deregulation. While the co-targeting of glycolysis and glutaminolysis is not a novel concept, the difficulty of using clinically relevant agents targeting such major metabolic pathways has dampened the efforts to translate pre-clinical studies. Our approach takes advantage of clinically approved molecules that are standards of

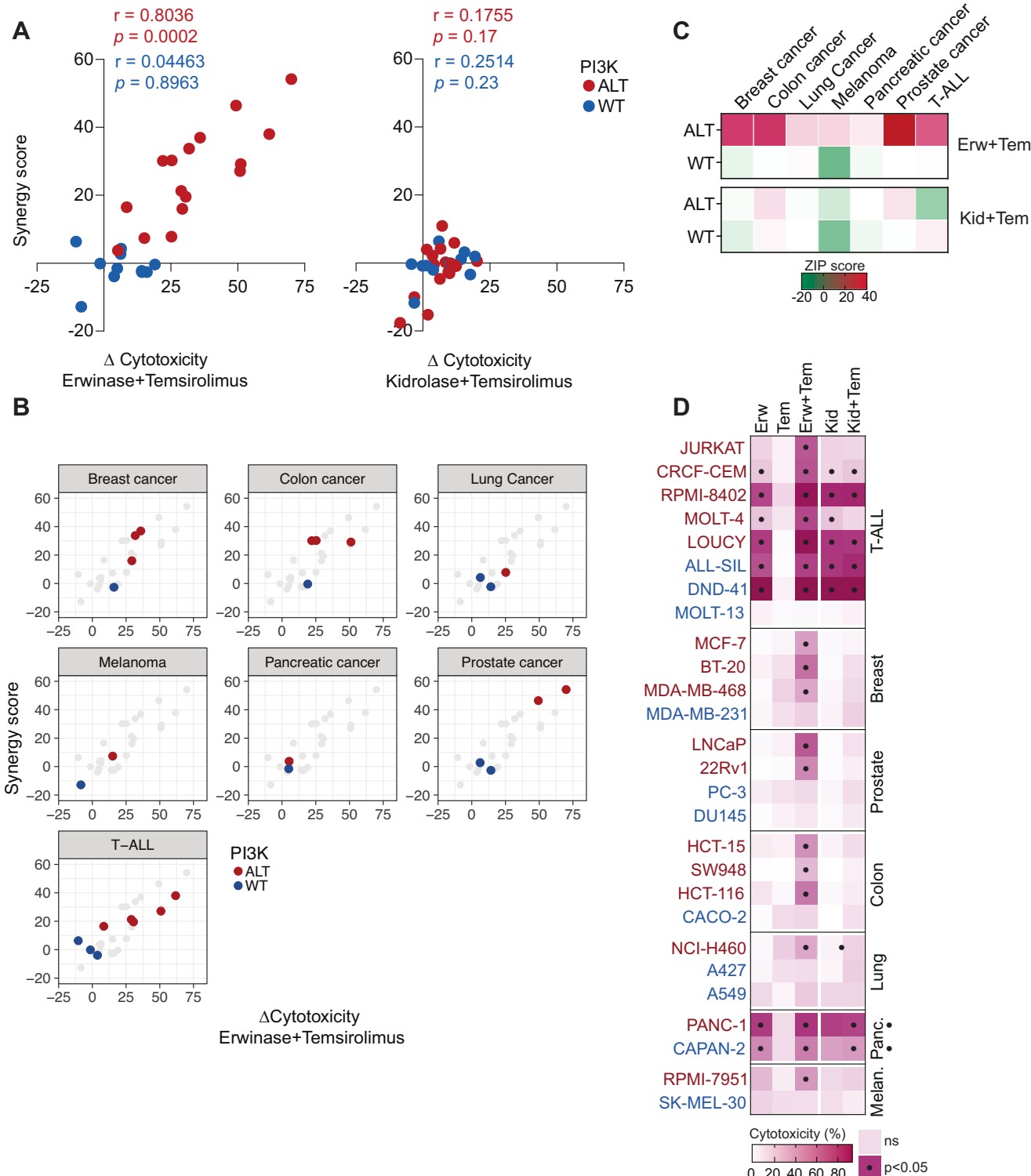

**Fig. 7 | PI3K-driven cancers are sensitive to the erwinase-temsirolimus combination.** **A** Pearson correlation between the difference of cytotoxicity of L-asparaginase (either erwinase – left or kidrolase – right) and temsirolimus minus the effect of L-asparaginase alone (Δ cytotoxicity) and the ZIP synergy score of the combination. **B** Pearson correlation between the Δ cytotoxicity and the ZIP synergy score of erwinase-temsirolimus stratified by cancer type. **C** ZIP synergy score of the combination of erwinase/temsirolimus (top) or kidrolase/temsirolimus (bottom) in the tested cancer models. **D** Cytotoxicity of erwinase (Erw), temsirolimus (Tem), kidrolase (Kid), or their combination on the broad array of either wild-type PI3K (blue) or PI3K-altered (red) solid cancer cell lines. Significative differences in cell viability relative to control are indicated by a black dot. A two-way ANOVA test adjusted for multiple comparisons (Dunn–Šidák correction) was computed.

cancer therapy. We emphasize the intricate connection between PI3K signaling and glucose consumption. While PI3K inhibitors fail to demonstrate cytotoxic effects, the use of small-molecule inhibitors of downstream effectors such as mTOR-targeting compounds is promising. Temsirolimus has proven its safety and demonstrated efficacies in various models and has been registered to treat solid cancers[35]. Erwinase, an L-asparaginase that also exhibits an L-glutaminase activity[29,36], is widely used in clinics to exploit the

vulnerability of leukemic blasts to extracellular pools of asparagine. L-asparaginases have demonstrated strong efficacy in the treatment of lymphoid malignancies, despite toxicities and resistance. A major predictor of L-asparaginase response is the methylation status of the *ASNS* regulatory elements[28,37]. *ASNS* encodes the asparagine synthetase, the enzyme catalyzing the de novo synthesis of asparagine, and is frequently silenced by hypermethylation in leukemia. This unique feature constitutes the rationale for the treatment of these hematological malignancies by recombinant L-asparaginases that degrade the extracellular asparagine, exploiting the inability of the blasts to reconstitute their pools. As a result, some patients may respond directly to L-asparaginases – kidrolase or erwinase, irrespective of their PI3K status. A recent study suggested that alterations of PI3K signaling may constitute a resisting mechanism to L-asparaginase in cell line models of T-ALL[38]. Yet, the authors do not mention the nature of the enzymes used in their study. In line with our data, they suggest that Akt targeting may re-sensitize resisting blasts to asparagine limitation. The methylation status of *ASNS* and the sensitivity to L-asparaginases is less reported in solid tumors. Yet, the preponderant role of asparagine and the reliance on the tumor cell-intrinsic ASNS activity has been reported in breast, prostate, melanoma, and pancreatic cancer models[39,40].

Our data demonstrate that the efficacy of erwinase resides in its glutaminase activity in PI3K-altered cancer cells, as kidrolase, another L-asparaginase lacking a robust affinity to glutamine, fails to demonstrate any cytotoxic effect. Of note, kidrolase provokes a transient reduction in glutamine pools in vivo, likely due to limited glutaminase activity and cell adaptation to the asparagine deprivation[41,42]. Similar glutamine targeting therapies have been underlined, usually combined with anti-mitochondrial agents, to exploit the dependence of cancer cells on these metabolic circuitries[43,44]. Venetoclax, an inhibitor of the BCL2 family, has been approved for patients with chronic lymphocytic and acute myeloid leukemias[45,46] and has been reported to be clinically active in patients with R/R T-ALL combined with salvage therapy[47–51]. We highlighted the efficacy and safety of the ET combination in five T-ALL/T-LL patients, three of whom also received venetoclax alongside the ET regimen. Given the efficacy of venetoclax on R/R T-ALL/T-LL, its potential additive effect when combined with ET could represent new therapeutic opportunities and requires further investigation. Hence, the elucidation of the metabolic liabilities of leukemic blasts will pave the way for the elaboration of novel targeting therapies combining anti-metabolite intervention with targeted pathways like mTOR.

Alterations of PI3K signaling are frequently observed in tumors, beyond hematological malignancies, as in breast, prostate, lung, colon, pancreas, or skin cancers. The oncogenic activation of PI3K signaling provides an unrepressed source of ATP to cancer cells through glycolysis to cope with their elevated energy demand. Hence, major efforts have been engaged to develop clinical-grade molecules that efficiently target PI3K signaling in cancer. Yet, only a few agents, including temsirolimus and everolimus, have positively demonstrated efficacy in clinical trials. These progresses have been limited by the toxicities and drug resistance profiles of PI3K inhibitors[4,35]. Novel PI3K isoform-specific molecules have recently been developed. Recent studies emphasized the targeting of PI3Kγ as a promising anti-leukemic strategy in models of acute myeloid leukemia[52,53]. PI3Kδ inhibitor idelalisib received approval for the treatment of relapsed B cell malignancies like chronic lymphocytic leukemia, follicular non-Hodgkin lymphoma, and small lymphocytic lymphoma, exploiting a specific profile restricted to leukocytes[54]. Another PI3Kδ inhibitor, umbralisib, and pan-class I PI3K inhibitor copanlisib were approved for the same indications. Recently, PI3Kα inhibitor alpelisib was approved in combination with fulvestrant for advanced/metastatic *PIK3CA*-mutated HR-positive HER2-negative breast cancer in postmenopausal women and men. In colorectal cancer, oncogenic activation of *PIK3CA* polarizes cell metabolism towards glutamine mobilization, hence constituting a putative therapeutic approach. Our data show that several PI3K-altered solid cancers are sensitive to the combination of erwinase and temsirolimus. While in-depth investigations are required to validate these results in pre-clinical settings, our study provides a strong rationale for therapeutic intervention targeting oncogenic-associated metabolic vulnerabilities in a larger array of tumors presenting PI3K alterations.

## Methods

All the experimentations were performed in compliance with the relevant ethical regulations of the GRAALL and FRALLE study groups, and the INEM IRB CEEA34 (APAFIS #8853-2017020814103710, #20996-2018021311224302, #2023121313089170).

### Patients

Patients included in this study were enrolled in the GRAALL-2003 (#NCT00222027[55]), GRAALL-2005 (#NCT00327678[56]), and FRALLE-2000[15] clinical trials. Patients treated with the combination of erwinase and temsirolimus were enrolled in the ALL-TARGET Observatory registry of relapsed/refractory T-cell acute lymphoblastic leukemia (ALL-TARGETOBS, #NCT05832125). All five patients had previously undergone front-line therapy according to the standard of care protocols in effect at their time of diagnosis (detailed in Supplemental Table 2). The combination of erwinase and temsirolimus was administered in an off-label setting for these patients. The ALL-TARGET Observatory (#NCT05832125) initiated by the GRAALL (Group for Research on Adult Acute Lymphoblastic Leukemia) is a real-world data registry that collects clinical data from patients with relapsed/refractory T-ALL for whom biological characterization, including oncogenetic, phenotypic, and in some cases functional analysis data, is available at diagnosis or relapse. The primary endpoint is the overall response rate with a focus on those who received targeted therapeutic option as a salvage therapy. Since the ALL-TARGET Observatory was initially designed for adult patients, four of the five patients were directly enrolled in this registry. The pediatric patient (T-LL-244) required a distinct management pathway, with the combination therapy of erwinase and temsirolimus being approved through a specialist committee review. This patient's clinical and biological data were incorporated into a dedicated database, following both parental written consent and specific institutional authorization (IRB #DC-2015-2473, Montpellier University Hospital). All sample collection and analyses were performed in compliance with the Declaration of Helsinki principles and received approval from the participating institutions review boards. Informed consent to collect and publish patient information was obtained from all adult subjects, while parental/guardian consent was obtained for the pediatric patient. One patient (T-LL-242), who was initially diagnosed at the age of 17, was 18 years old at the time of progression/relapse and therefore met the age-based inclusion criteria for the ALL-TARGET Observatory, enabling their enrollment in the registry.

### Next-generation sequencing, copy number, and molecular analyses

Targeted whole-exon sequencing of 105 genes relevant to hematological malignancies was performed using a custom Nextera XT gene panel (Illumina, San Diego, CA) as previously described[34,57]. Eligible samples were screened for fusion transcripts and oncogene ectopic expression as reported[14,58]. Copy number variants were assessed by array-based comparative genomic hybridization[14,34] and multiplex ligation-dependent probe amplification (MLPA) technique with the SALSA-MLPA P383 T-ALL probe mix (MRC-Holland, Amsterdam, Netherlands) kit.

### Mice and in vivo experimentation

NOD.Cg-*Prkdc*^SCID *Il2rg*^tm1Wjl/SzJ (NSG) mice were purchased from Charles River and maintained in a specific and opportunistic pathogen-

free LEAT Animal facility at the Institut Necker-Enfants Malades. Patient-derived xenografts (PDX) were generated as previously described[57]. Briefly, $10^6$ fresh viable primary blasts were xenografted by intravenous retro-orbital injection in 6-week-old NSG mice. The leukemic burden in the blood was monitored weekly by flow cytometry as the percentage of FSC[hi], hCD7+ hCD45+ cells. For in vivo experimentation, mice were segregated into treatment arms when the leukemic load reached 0.5% to 5% (5 mice per arm) and treated for two cycles of five consecutive days. Telaglenastat (CB-839) was given by oral gavage (200 mg/kg) twice daily. Temsirolimus was administrated daily at 10 mg/kg by intraperitoneal injection. Erwinase or kidrolase were given at 2 UI/g by intravenous retro-orbital injection every two days. Mice were euthanized when terminally ill, as evidenced by a blood infiltration >80%, severe dyspnea or frailty caused by leukemic dissemination in the mediastinum or vital organs, in accordance to the IRB approval. The maximal tumor burden of 90% blood infiltration was not exceeded. For ex vivo experimentation, bone marrows from tibiae, hips, femurs, and vertebrae were collected and blasts purified by Ficoll followed by hCD45+ magnetic sorting (Miltenyi). The blast purity was checked by flow cytometry (percentage of FSC[hi], hCD7+ hCD45+ cells) before experimentation. All the samples used contained ≥90% human blasts.

## Cell culture

The T-ALL cell lines JURKAT, DND-41, ALL-SIL, RPMI-8402, CCRF-CEM, LOUCY, PEER, MOLT-4, and MOLT-13 were obtained and authenticated from ATCC and DSMZ. The breast (MCF-7, BT-20, MDA-MB-468 and MDA-MB-231), prostate (LNCaP, 22Rv1, PC-3, DU145), colon (HCT-15, SW948, HCT-116 and CACO-2), lung (NCI-H460, A427 and A549), pancreatic (PANC-1, CAPAN-2) and melanoma (RPMI-7951 and SK-MEL-30) cancer cell lines were obtained from DSMZ. Cell lines were grown in RPMI-1640, DMEM, or DMEM-F12 according to the supplier information, supplemented with 50 μg/mL streptomycin, 50 UI penicillin, 4 mM L-glutamine (complete medium), and 10% fetal bovine serum (Gibco). Ex vivo cultures of PDX were achieved in a complete medium, supplemented with 50 ng/ml human stem cell factor, 20 ng/ml hFLT3-L, 10 ng/ml hIL-7, and 20 nM insulin (Miltenyi Biotec, Bergisch Gladbach, Germany) and 20% fetal bovine serum. When mentioned, glucose-free (Gibco #11879020) or glutamine-free (Gibco #21870092) RPMI-1640 was used, supplemented as stated above, without the addition of glutamine for the glutamine-free condition. Cultures were maintained at 37 °C in a humidified atmosphere containing 5% $CO_2$.

## Inhibitors and drugs

Wortmannin was purchased from Sigma (#12-338). Temsirolimus was obtained from Abmole (#M3722). Telaglenastat (CB-839) was purchased from MedChemExpress (#HY-12248). Kidrolase was obtained from the Necker Hospital pharmacy service. Erwinase was collected from the residual volumes of infusion bags administrated to patients at the Necker Hospital.

## Cell proliferation and viability

Cell proliferation and viability were monitored by flow cytometry. For proliferation assays, cells were stained with CellTrace™ Violet (BioLegend) before culture. Cells were washed in ice-cold PBS and collected by centrifugation (5 min, 350x g, 4 °C) before staining with Annexin V and propidium iodide in Annexin V binding buffer (BioLegend).

## Glucose consumption

Glucose uptake was evaluated using the Glucose Uptake Cell-Based Assay Kit (Cayman Chemical #600470) and the supplier protocol. Briefly, cells were seeded and treated as requested before a 30-minute incubation with 2-NBDG diluted in a glucose-free medium (100 μg/ml). Cells were then collected by centrifugation (5 min, 350 x g, 4 °C) and washed in an ice-cold Cell-based Assay Buffer with propidium iodide before immediate analysis by flow cytometry.

## Phosflow

Cells were washed in ice-cold PBS and collected by centrifugation (5 min, 350x g, 4 °C). Cells were incubated with anti-hCD45 V500 (HI30, BD Biosciences) and Zombie NIR (BioLegend) on ice for 15 min in ice-cold PBS and washed, fixed for 10 min at 4 °C and (Fixation Buffer #420801, BioLegend) and washed, then permeabilized 30 min at 4 °C (PermBuffer III®, BD Biosciences). After two washes with ice-cold PBS-BSA 0.5%, cells were incubated with anti-pS473-Akt PE (BD Phosflow™ M89-61, BD Biosciences), pS235/236-S6 V450 (BD Phosflow™ N7-548, BD Biosciences) and pT36/45-4E-BP1 Alexa Fluor® 647 (BD Phosflow™ M31-16, BD Biosciences). Fold changes were calculated from relative fluorescence intensities in viable hCD45+ blasts.

## RNA sequencing and data analysis

155 primary samples were analyzed by RNA sequencing. RNA eligibility was determined by measurement on a BioAnalyzer (RIN ≥ 8, Agilent). Libraries were prepared using the SureSelect XT HS2 kit (Agilent) and sequenced on a NovaSeq sequencer (Illumina). Reads were trimmed using Agent v2.0.5 with 'trim -v2' arguments and aligned to the GRCh38 genome assembly using STAR v2.7.9 with default parameters. before deduplication using Agent with 'locatit mbc -i -R' arguments. Gene-level counting was done using subread featureCounts v2.0.0 with -O argument against Ensembl release 95 gtf file. Analyses were carried out on a subset of 4,871 metabolic genes obtained from the Reactome Pathway database (ID R-HSA-1430728). Differential expression analyses were done with DESeq2 on R[59]. Pre-filtering was performed as recommended by DESeq2 documentation (conservation of features with at least 10 reads for the smallest group of samples (ALT, $n = 33$). 2,141 metabolic genes were preserved in the dataset and subjected to differential expression analysis (ALT vs WT). 243 differentially expressed genes (DEG) were identified. Post-analysis filtering was applied to select the most variable genes (absolute log2 fold change > 1 and adjusted $p$ value < 0.05). 75/243 genes were identified and used for hierarchical clustering and enrichment analyses with Enrichr[60]. Gene set enrichment analyses (GSEA) using the expression of the 2,141 metabolic genes were computed using the Broad Institute GSEA software[61].

## Targeted LC-MS metabolomics analyses

For metabolomic analysis, the extraction solution was composed of 50% methanol, 30% acetonitrile (ACN), and 20% water. The volume of the extraction solution was adjusted to cell number (1 ml per 1E7 cells). After the addition of extraction solution, samples were vortexed for 5 min at 4 °C and centrifuged at 16,000 g for 15 min at 4 °C. The supernatants were collected and stored at −80 °C until analysis. LC/MS analyses were conducted on a QExactive Plus Orbitrap mass spectrometer equipped with an Ion Max source and a HESI II probe coupled to a Dionex UltiMate 3000 uHPLC system (Thermo). External mass calibration was performed using a standard calibration mixture every seven days, as recommended by the manufacturer. The 5 μl samples were injected onto a ZIC-pHILIC column (150 mm × 2.1 mm; i.d. 5 μm) with a guard column (20 mm × 2.1 mm; i.d. 5 μm) (Millipore) for LC separation. Buffer A was 20 mM ammonium carbonate, 0.1% ammonium hydroxide (pH 9.2), and buffer B was ACN. The chromatographic gradient was run at a flow rate of 0.200 μl min⁻¹ as follows: 0–20 min, linear gradient from 80% to 20% of buffer B; 20–20.5 min, linear gradient from 20% to 80% of buffer B; 20.5–28 min, 80% buffer B. The mass spectrometer was operated in full scan, polarity switching mode with the spray voltage set to 2.5 kV and the heated capillary held at 320 °C. The sheath gas flow was set to 20 units, the auxiliary gas flow to 5 units, and the sweep gas flow to 0 units. The metabolites were detected across a mass range of

75–1,000 $m/z$ at a resolution of 35,000 (at 200 $m/z$) with the automatic gain control target at $10^6$ and the maximum injection time at 250 ms. Lock masses were used to ensure mass accuracy below 5 ppm. Data were acquired with Thermo Xcalibur software (Thermo). The peak areas of metabolites were determined using Thermo TraceFinder software (Thermo), identified by the exact mass of each singly charged ion and by the known retention time on the HPLC column. Enrichment analyses were carried out using MetaboAnalyst 5.0. Metabolite concentrations were normalized by sum, log-transformed, and auto-scaled before statistical analysis.

### Tracing metabolomics with U-[$^{13}$C]-Glucose and U-[$^{13}$C]-Glutamine

T-ALL PDX blasts were purified from mice and cultured in a glucose-free (Gibco #11879020) medium, supplemented with 50 ng/ml human stem cell factor, 20 ng/ml hFLT3-L, 10 ng/ml hIL-7, and 20 nM insulin (Miltenyi Biotec, Bergisch Gladbach, Germany) and 20% fetal bovine serum for 24 h. After the deprivation period, U-[$^{13}$C]-Glucose (1 mM) or U-[$^{13}$C]-Glutamine (2 mM) (Cambridge Isotope) was added to the culture media. A total of 5 million viable cells were collected per timepoint (1 h, 4 h, 24 h) in three replicates per sample, washed twice in pre-chilled PBS by centrifugation at 4 °C, snap-frozen and stored at −80 °C for further LC-MS–based metabolomic analyses.

### Statistical analyses

Normality tests were applied to determine if the datasets were eligible for either parametric or nonparametric tests. Statistical analyses were performed with Student *t*-test, ANOVA, or Fisher test according to the dataset nature by using GraphPad Prism 8 software (GraphPad Software, Inc., San Diego, CA, United States) and R (v. 4.3.1). Pearson correlations were computed with GraphPad Prism 8. Zero interaction potency (ZIP) synergy scores were computed using the R synergy-finder package[62]. Survival analyses were carried out with R with a log-rank test. Overall survival (OS) and event-free survival (EFS) were respectively defined as the time from diagnosis to death or the occurrence of an adverse event, censoring patients alive at the last follow-up. Cumulative incidence of relapse (CIR) was defined as the time from complete remission to relapse, censoring patients alive and in continuous complete remission at the time of the last follow-up. Death in complete remission was considered a competing event for CIR. Outcome comparisons were performed using a univariate Fine & Gray test for CIR and a Cox regression analysis for OS. The following symbols were used to indicate significant differences: ns, $p > 0.05$; *, $p < 0.05$; **, $p < 0.01$; ***, $p < 0.001$.

### Reporting summary

Further information on research design is available in the Nature Portfolio Reporting Summary linked to this article.

## Data availability

The RNA raw sequencing data in the form of FASTQ files, have been deposited in the European Genome-phenome Archive (EGAD00001010273). Additional patient data generated in this study are not publicly available due to information that could compromise patient privacy or consent but are available upon reasonable request from the corresponding authors. Access to these data must be requested to the corresponding authors and submitted to the study groups. There is no time limit to data availability and an answer will be provided within a month. Detailed methodology regarding metabolite identification in targeted metabolomics can be obtained from Dr. Nemazanyy (ivan.nemazanyy@inserm.fr, Platform for Metabolic Analyses, Structure Fédérative de Recherche Necker, INSERM US24, CNRS UAR3633, Université Paris Cité, Paris, France). Source data used to generate the figures are provided as a Source Data file. Source data are provided with this paper.

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

## Acknowledgements

The authors thank all the participants of the GRAALL-2003, GRAALL-2005, and FRALLE2000 study groups, the SFCE and the investigators of the 16 SFCE centers involved in the collection and provision of data and patient samples, and V. Lhéritier for collection of clinical data. The authors also thank the Structure Fédérative de Recherche Necker LEAT core facility. This work was supported by grants to the Necker laboratory from the Association pour la Recherche contre le Cancer (Equipe labellisée), Ligue Nationale Contre le Cancer (Equipe labellisée), Institut National du Cancer PRT-K 18-071 and the Agence Nationale de la Recherche (Institut THEMA Saint-Louis, ANR-23-IAHU-0005). The GRAALL was supported by grants P0200701 and P030425/AOM03081 from the Programme Hospitalier de Recherche Clinique, Ministère de l'Emploi et de la Solidarité, France and the Swiss Federal Government in Switzerland. Samples were collected and processed by the AP-HP "Direction de Recherche Clinique" Tumor Bank at Necker-Enfants Malades. G. P. A was supported by the Fondation de France (FDF) and INSERM. M. S. was supported by « Action Leucémie » and "Ligue contre le Cancer".

## Author contributions

Design of the study: G.P.A., M.S., P.R., O.H., V.A. Experimentation: G.P.A., M.S., G.H., J.D., I.N. Biological data acquisition and analysis: G.P.A., M.S., G.H., J.D., I.N. Clinical data acquisition and analysis: M.S., A.C.H., E.L., A.M., A.T., M.E.D., N.B., H.D., P.R., V.A. Funding acquisition: G.P.A., M.S., P.R., O.H., V.A. Manuscript writing and correction: G.P.A., M.S., P.R., O.H., V.A.

## Competing interests

The authors declare no competing interests.
