## [Transparent Peer Review file · Nature Communications]

A metabolic synthetic lethality of phosphoinositide 3-kinase-driven cancer

Corresponding Author: Professor Vahid Asnafi

Version 0:

Reviewer comments:

Reviewer #1

(Remarks to the Author)

In this manuscript, Andrieu et al apply to investigate metabolic vulnerabilities of PI3K-driven T-ALL. They begin by showing that PI3K pathway mutations are associated with inferior prognosis in T-ALL patients, and with a gene expression signature suggesting increased glycolysis, and provide additional evidence that this is a functional effect based on increased glucose uptake by PI3K altered T-ALL cell lines. These cell lines are intolerant of glucose deprivation, as opposed to the PI3K WT lines, however this was not the case for PI3K altered PDX models. Metabolomics analysis suggested that PI3K altered (but not WT) cases consume glutamine when glucose is lacking, a very interesting observation. PI3K-altered cells upregulate glutamine uptake in the absence of glucose, and pharmacologic glutaminase inhibition was toxic in combination with glucose deprivation or mTOR inhibition. mTOR inhibition was also toxic in combination with Erwinase, a form of asparaginase with glutaminase activity, but not with a glutaminase-free asparaginase, both in vitro and in vivo in preclinical models. Strikingly, the authors also treated 5 patients with relapsed/refractory T-ALL with erwinase and the mTOR inhibitor temsirolimus, and all went into at least a transient complete response (although most also received venetoclax, complicating interpretation). This combination is also shown to have activity in other tumor types with PI3K activation. Overall there are key interesting parts to this manuscript, but I also have a number of concerns that need to be addressed:

1. References or details of therapy used on the clinical trials from which these samples were obtained are lacking. If the trials are not published, can a summary of therapy be added to the methods or the supplement?
2. I am surprised that the authors are calling MOLT13 a PI3K signaling pathway wild-type cell (Fig 3a) because these have been described by different groups as being p473-AKT high and PTEN-null (PMID: 17873882 and 29799846). What is the data that the authors are using to classify the different cell lines in Figure 3 as PI3K WT versus altered? Can the authors provide biochemical evidence (i.e., the assays in Fig 2A) to support that PI3K signaling correlates with the WT vs altered PI3K pathway calls, and can they also provide at least one (ideally two) more wild-type cell lines to ensure the two WT cell lines here are not outliers? STR genotyping should also be performed to rule out the possibility of misidentified cell lines.
3. Similar to the above comment, what is the data the authors are using to call PDX models as PI3K WT or altered in Fig 3C?
4. Line 184, I do not understand why glucose limitation should induce an amino acid response pathway. The authors should explain what amino acid response they are talking about and why this should be induced by glucose limitation, or this statement could be removed as it is not essential for this story.
5. What is the link between glutamine and arginine biosynthesis that Fig 4C purports to show? To my knowledge, the major route for arginine biosynthesis is arginosuccinate lyase, I do not understand how this links to glutamine. This needs explanation. If this is a prediction based on some pathway prediction algorithm, I would suggest this is too speculative to include here, unless this can be substantiated with hard data such as metabolic flux analysis (more on that below).
6. Changes in steady-state metabolite concentrations cannot be used to infer changes in metabolic flux, as a change could indicate increased or decreased flux through the pathway. It is also not clear to this reviewer that the data supports the conclusion that glutamine is being used to support the TCA, it could be used in other processes (both of these claims are made in lines 198-199). Glutamine metabolic flux analysis should be performed to strengthen these conclusions.

7. Fig. 5D, right panel: I am not familiar with the ZIP score assessment of synergy and I suspect this will also be a problem for at least some readers, this should be explained. Alternatively, the Bliss panel on the left should be sufficient.

8. The degree of response in each patient is difficult to decipher from Fig 6A. I do appreciate that the authors are trying to provide as much information as possible, but this figure is very difficult to decipher. Can a separate plot of the percent blasts in the bone marrow (by morphology or flow or MRD) pre- and post-chemotherapy, at least for the patients who had extensive bone marrow involvement?

9. The discussion of the Warburg effect in the introduction and in the first paragraph of the discussion seems to miss what I do believe is the contemporary explanation for this phenomenon, which is that its main role is not ATP production but rather to provide substrates for biomass production, as reviewed in ref #17 cited by the authors. Another insightful discussion is PMID: 32694689. This first part of the discussion and also the 2nd paragraph of the introduction should be rewritten to better capture this contemporary understanding.

Minor:

1. "PI3KS" is a nonstandard abbreviation that confused me more than once, recommend spelling out PI3K signaling. PI3KS-ALT is another nonstandard abbreviation that is not defined. I would prefer to see the authors avoid nonstandard abbreviations as much as possible.

2. I have a number of requested edits to the language to avoid the impression that imprecise terms are being used in several parts of the manuscript:

- The outcomes in Fig 1B do not look "dismal" to me, this is not too bad for a T-ALL cohort that includes a lot of adults. Suggest using a less ominous term in line 128, perhaps "inferior".

- The title to the section on lines 178-179 talks about a "vulnerability to glutamine metabolism", however whether or not glutamine is a vulnerability is not addressed in this section of the results. The title should be changed to reflect what is being shown in this section of the results.

- What the authors mean by "ET association" on line 262 needs to be defined.

Reviewer #2

(Remarks to the Author)

The current manuscript by Guillaume P. Andrieu et al. describes a phenomenon that T-ALL cells with genetic alteration in PI3K signaling pathway are more susceptible to glutamine deprivation when glucose is limited or when mTOR pathway is inhibited. Therefore, they designed a combination therapy composed of Erwinase and Temozolomide to block the utilization of glutamine and glucose simultaneously to target T-ALL with mutant PI3K pathway. There are several concerns regarding the novelty, misunderstood metabolic concepts and experiment design of current manuscript.

Major concerns:

1. It has been reported that several types of cancer cells are prone to utilize glutamine when glucose is limited, especially PTEN mutant cells. Therefore, the novelty of current manuscript is compromised.

2. The authors use two different types of Asparaginase, one with glutaminase activity (Erwinase). In vitro studies show that Erwinase is more toxic to leukemic cells when combined with Temozolomide, but whether this toxicity depends on the asparaginase activity? Especially for the in vivo studies, whether asparaginase activity is critical for killing leukemic cells for the combination therapy? In vivo study using CB-839 combined with Temozolomide would be a good experiment to test the authors' hypothesis regarding the synthetic lethality combining inhibition of glucose and glutamine utilization.

3. The patient results from Fig.6 are the highlight of current manuscript. But when carefully examined the reported results, in the presence of Venetoclax (Ven) can patients be disease free. Therefore, it seems like the Erw/Temo therapy needs to be combined with Ven to exhibit strong anti-leukemic effects, which does not support the authors' hypothesis.

4. The authors use glucose uptake as a parameter for glucose utilization. Although glucose uptake sometimes may correlate with glucose utilization, they are two different concepts. Metabolomics studies are required to track glucose flow to assess glucose utilization. And there are not such studies in the current manuscript.

5. Similarly, one cannot conclude that glutaminolysis is activated/inhibited from a metabolic screen showing decreased/elevated glutamine/glutamate levels. Assessing the glutamine flow is required to get such conclusion. And again, there are not such studies in the current manuscript.

Minor concerns:

There are not enough experimental details in the Figure legends/methods.

Version 1:

Reviewer comments:

Reviewer #1

(Remarks to the Author)

The authors have done a very nice job with revisions and addressed all of my concerns.

Reviewer #2

(Remarks to the Author)

The authors have addressed the reviewer's concerns.

REVIEWER COMMENTS

Reviewer #1 (Remarks to the Author):

In this manuscript, Andrieu et al apply to investigate metabolic vulnerabilities of PI3K-driven T-ALL. They begin by showing that PI3K pathway mutations are associated with inferior prognosis in T-ALL patients, and with a gene expression signature suggesting increased glycolysis, and provide additional evidence that this is a functional effect based on increased glucose uptake by PI3K altered T-ALL cell lines. These cell lines are intolerant of glucose deprivation, as opposed to the PI3K WT lines, however this was not the case for PI3K altered PDX models. Metabolomics analysis suggested that PI3K altered (but not WT) cases consume glutamine when glucose is lacking, a very interesting observation. PI3K-altered cells upregulate glutamine uptake in the absence of glucose, and pharmacologic glutaminase inhibition was toxic in combination with glucose deprivation or mTOR inhibition. mTOR inhibition was also toxic in combination with Erwinase, a form of asparaginase with glutaminase activity, but not with a glutaminase-free asparaginase, both in vitro and in vivo in preclinical models. Strikingly, the authors also treated 5 patients with relapsed/refractory T-ALL with erwinase and the mTOR inhibitor temsirolimus, and all went into at least a transient complete response (although most also received venetoclax, complicating interpretation). This combination is also shown to have activity in other tumor types with PI3K activation. Overall there are key interesting parts to this manuscript, but I also have a number of concerns that need to be addressed:

1. References or details of therapy used on the clinical trials from which these samples were obtained are lacking. If the trials are not published, can a summary of therapy be added to the methods or the supplement?

The patients were previously treated with various lines of chemotherapy, including protocols for T-ALL and T-LL (both adult and pediatric), such as FRALLE2000, CAALLF01, GRAALL2014, GRAALL-LYSA LLO3, COOPRALL, and NECTAR. In response to the reviewer's request (Q.8), we have simplified Figure 6 and provided information about the different treatment protocols in the updated Supplemental Table 2. Additionally, detailed descriptions of these protocols have been included in the Supplemental Methods.

2. I am surprised that the authors are calling MOLT13 a PI3K signaling pathway wild-type cell (Fig 3a) because these have been described by different groups as being p473-AKT high and PTEN-null (PMID: 17873882 and 29799846). What is the data that the authors are using to classify the different cell lines in Figure 3 as PI3K WT versus altered? Can the authors provide biochemical evidence (i.e., the assays in Fig 2A) to support that PI3K signaling correlates with the WT vs altered PI3K pathway calls, and can they also provide at least one (ideally two) more wild-type cell lines to ensure the two WT cell lines here are not outliers? STR genotyping should also be performed to rule out the possibility of misidentified cell lines.

We think that there might be confusion between MOLT-3 and MOLT-13 for the reviewer. MOLT-13 is a PI3Kwt cell line and is reported as such on DepMap (DepMap ID: ACH-000795). In the study PMID 17873882 cited by the reviewer, this is mention of MOLT-3, a PTEN-null

cell line that we use as such. Here is a capture of the related article to clarify this point (PMID 17873882 Fig1b).

Furthermore, in the study PMID 29799846, no data indicate a mutation for PTEN in the MOLT-13 cell line. It is shown that this cell line expresses the different subunits of class I PI3K (p110a,b,g,d). It is widely reported that basal activation of the PI3K pathway can occur and be detected by pAkt without any mutation of the core members. Additionally, epigenetic deregulations of the PI3K pathway have also been published (Tottone et al Blood Cancer Discov 2021).

As suggested by the Reviewer, we now have added two PI3K wt cell lines DND-41 (Palomero et al 2007) and PEER (Zuurbier et al 2012) to Figures 3A and 3B, with similar glucose uptake rates and survival behavior facing glucose limitation than the previous two other PI3K-wt cell lines.

3. Similar to the above comment, what is the data the authors are using to call PDX models as PI3K WT or altered in Fig 3C?

PI3K signaling alterations were defined based on patient and PDX molecular analysis by whole exome targeted sequencing and copy number analysis by MLPA and array-CGH (as described in our recent manuscript reporting the impact of gene alterations in both adult and pediatric T-ALL (Simonin et al. Blood 2024, PMID: 38848537). The conservation of these alterations from patient to PDX are described in Supplementary Figure 2.

4. Line 184, I do not understand why glucose limitation should induce an amino acid response pathway. The authors should explain what amino acid response they are talking about and why this should be induced by glucose limitation, or this statement could be removed as it is not essential for this story.

While it has been proposed that the amino acid response pathway is engaged upon nutrient deprivation including glucose, we agree that this sentence is speculative and we cannot

provide clear data to support it. As suggested by the reviewer, we have removed it, as not essential to the message of the paper.

5. What is the link between glutamine and arginine biosynthesis that Fig 4C purports to show? To my knowledge, the major route for arginine biosynthesis is arginosuccinate lyase, I do not understand how this links to glutamine. This needs explanation. If this is a prediction based on some pathway prediction algorithm, I would suggest this is too speculative to include here, unless this can be substantiated with hard data such as metabolic flux analysis (more on that below).

We thank the reviewer for this point. Indeed, the networks presented initially in Fig 4C are produced by algorithms. Tracing experiments realized for the revision of this study did not show active synthesis of arginine from glutamine. Hence, as suggested, we have removed this figure from the revised manuscript.

6. Changes in steady-state metabolite concentrations cannot be used to infer changes in metabolic flux, as a change could indicate increased or decreased flux through the pathway. It is also not clear to this reviewer that the data supports the conclusion that glutamine is being used to support the TCA, it could be used in other processes (both of these claims are made in lines 198-199). Glutamine metabolic flux analysis should be performed to strengthen these conclusions.

We thank the reviewer for this important point. To address this point, we have conducted a metabolic flux analysis using U[¹³C]-glutamine on six independent PDX (3 wt, 3 alt) *ex vivo* in triplicate. The data are now included in Figure 4C and Supplementary Figure 4. We now provide clear evidence supporting that glutamine is mobilized to support the TCA cycle upon glucose limitation in PI3K-altered T-ALL PDX. Regarding other metabolic processes, we report no induction of reductive carboxylation of glutamine or *de novo* asparagine synthesis, hence indicating that glutaminolysis is the main metabolic pathway induced in these conditions. Overall, these novel data validate that targeting glucose and glutamine metabolism is synthetically lethal in PI3K-driven leukemia.

7. Fig. 5D, right panel: I am not familiar with the ZIP score assessment of synergy and I suspect this will also be a problem for at least some readers, this should be explained. Alternatively, the Bliss panel on the left should be sufficient.

The Zero interaction potency (ZIP) model captures the drug interaction relationships by comparing the change in the potency (effect at a certain dose level) of the dose-response curves between individual drugs and their combinations. ZIP assumes that two non-interacting drugs are expected to incur minimal changes in their dose-response curves. The formulation of the model is described in Yadav *et al* Comput Struct Biotechnol J 2015.

ZIP model takes advantage of both the Loewe additivity and the Bliss independence models, aiming at a systematic assessment of various types of drug interaction patterns that may arise in a high-throughput drug combination screening.

8. The degree of response in each patient is difficult to decipher from Fig 6A. I do appreciate that the authors are trying to provide as much information as possible, but this figure is very difficult to decipher. Can a separate plot of the percent blasts in the bone marrow (by morphology or flow or MRD) pre- and post-chemotherapy, at least for the patients who had extensive bone marrow involvement?

We thank the reviewer for helping us to improve the legibility of the figure. Figure 6 has been revised to provide better insights into the patients' outcomes, blast clearance in the bone marrow, and the responses assessed by imaging.

9. The discussion of the Warburg effect in the introduction and in the first paragraph of the discussion seems to miss what I do believe is the contemporary explanation for this phenomenon, which is that its main role is not ATP production but rather to provide substrates for biomass production, as reviewed in ref #17 cited by the authors. Another insightful discussion is PMID: 32694689. This first part of the discussion and also the 2nd paragraph of the introduction should be rewritten to better capture this contemporary understanding.

We thank the reviewer, as this suggestion has improved the quality of the mentioned sections. We have rewritten them accordingly to reflect the modern comprehension of the Warburg effect.

Minor:

1. "PI3KS" is a nonstandard abbreviation that confused me more than once, recommend spelling out PI3K signaling. PI3KS-ALT is another nonstandard abbreviation that is not defined. I would prefer to see the authors avoid nonstandard abbreviations as much as possible.

These nonstandard abbreviations have been changed throughout the manuscript.

2. I have a number of requested edits to the language to avoid the impression that imprecise terms are being used in several parts of the manuscript:

- The outcomes in Fig 1B do not look "dismal" to me, this is not too bad for a T-ALL cohort that includes a lot of adults. Suggest using a less ominous term in line 128, perhaps "inferior".

We agree with the reviewer and have changed the term accordingly (now line 139).

- The title to the section on lines 178-179 talks about a "vulnerability to glutamine metabolism", however whether or not glutamine is a vulnerability is not addressed in this section of the results. The title should be changed to reflect what is being shown in this section of the results.

We agree with the reviewer and have changed the title of this section to better reflect the results presented (now line 223).

- What the authors mean by “ET association” on line 262 needs to be defined. We have defined ‘ET’ as ‘Erwinase/Temsirolimus’ (now line 360).

Reviewer #2 (Remarks to the Author):

The current manuscript by Guillaume P. Andrieu et al. describes a phenomenon that T-ALL cells with genetic alteration in PI3K signaling pathway are more susceptible to glutamine deprivation when glucose is limited or when mTOR pathway is inhibited. Therefore, they designed a combination therapy composed of Erwinase and Temsirolimus to block the utilization of glutamine and glucose simultaneously to target T-ALL with mutant PI3K pathway. There are several concerns regarding the novelty, misunderstood metabolic concepts and experiment design of current manuscript.

Major concerns:

1. It has been reported that several types of cancer cells are prone to utilize glutamine when glucose is limited, especially PTEN mutant cells. Therefore, the novelty of current manuscript is compromised.

As we read the incipit and the first concern raised by this reviewer, we feel that the remark on the novelty of the results being compromised seems not fully justified and may reflect a misunderstanding of our main message. As largely cited and commented throughout the introduction and the discussion of our manuscript, the polarization of PI3K-dysregulated tumors towards glucose, and in some studies to glutamine, has been coined, notably in the cited references.

The novelty of our study lies in the discovery that glutamine metabolism serves as the unique metabolic salvage pathway for PI3K-driven leukemia, creating a metabolic synthetic lethality. Importantly, we demonstrate that this vulnerability can be clinically targeted, and we present clinical data from very high-risk patients (those with primary refractory disease or multiple relapses, with no remaining conventional therapeutic options) who exhibited strong responses to the novel targeted therapy—one that we are the first to propose and test in patients.

While we clearly say that the metabolic polarization of PI3K-driven leukemia is in line with the existing literature, we identify a metabolic vulnerability leading to a synthetic lethality and propose a novel clinical-grade combination that is promising and currently evaluated in a clinical observatory (NCT05832125). In addition, based on this solid observation made in multiple cancer models, we provide an efficient and clinically relevant strategy to counter it. This represents the novelty of our study, hereby reinforcing the perspective of our approach

beyond leukemia.

2. The authors use two different types of Asparaginase, one with glutaminase activity (Erwinase). *In vitro* studies show that Erwinase is more toxic to leukemic cells when combined with Temsirolimus, but whether this toxicity depends on the asparaginase activity? Especially for the *in vivo* studies, whether asparaginase activity is critical for killing leukemic cells for the combination therapy? *In vivo* study using CB-839 combined with Temsirolimus would be a good experiment to test the authors' hypothesis regarding the synthetic lethality combining inhibition of glucose and glutamine utilization.

This is an important point, and we thank the reviewer for allowing us to add new data that strengthens our main message. In the revised manuscript, we have added new *in vivo* data showing that anti-glutaminotics are essential to target the metabolic synthetic lethality of PI3K-driven leukemia. First, we now provide an *in vivo* comparison of erwinase vs kidrolase combined with temsirolimus. Importantly, only erwinase, the asparaginase with glutaminase activity significantly controls leukemia outgrowth *in vivo*. Second, the combination of temsirolimus with CB-839, a genuine inhibitor of GLS that prevents the anaplerotic flux had a significant anti-leukemic effect *in vivo*.

Altogether, these new data clearly show that the metabolic synthetic lethality of PI3K-driven leukemia is targetable by combining a mTOR inhibitor with glutamine-targeting intervention.

3. The patient results from Fig.6 are the highlight of current manuscript. But when carefully examined the reported results, in the presence of Venetoclax (Ven) can patients be disease free. Therefore, it seems like the Erw/Term therapy needs to be combined with Ven to exhibit strong anti-leukemic effects, which does not support the authors' hypothesis.

While the addition Venetoclax to the Erwinase-Temsirolimus combination may offer benefits (ref), two patients (UPNT-1402 and LLT-147) achieved complete responses with the combination of Erwinase and Temsirolimus. Thus, the T-ALL patient (UPNT-1402) became MRD-negative, and the T-LL patient (LLT-147) experienced a 75% reduction in mediastinal mass, demonstrating the efficacy of this regimen without Venetoclax. Notably, the patient (LBL-T 242) who initially responded to the Erwinase-Temsirolimus-Venetoclax combination experienced disease progression after Erwinase was discontinued due to supply limitations, despite continued treatment with Temsirolimus and Venetoclax. This highlights the crucial role of Erwinase in the chemotherapy regimen.

The use of venetoclax in these patients stemmed from the physician's practice over the past few years of adding venetoclax for relapsed or refractory cases, based on preliminary yet encouraging results from various studies. Importantly, these studies highlight that venetoclax tends to be more effective in immature T-ALL than in mature T-ALL (which describes the majority of our patients) due to differences in anti-apoptotic protein dependence related to the differentiation stage of the T-ALL clone (Chonghaile et al., Cancer Discovery 2014, PMID: 24994123). Importantly, patients with PI3K alterations are more commonly associated with a mature or cortical T-ALL phenotype, which is linked to BCL-XL dependence, rendering them more sensitive to navitoclax than venetoclax.

Overall, we suspect that venetoclax had limited efficacy in our study's patients due to their maturation arrest, which likely made them more sensitive to navitoclax (a BCL-XL inhibitor) rather than venetoclax (a BCL-2 inhibitor). Notably, venetoclax is expected to play a pivotal role in future T-ALL trials, both in frontline therapy and in relapsed cases. Discussions are currently ongoing regarding the addition of venetoclax in frontline treatment for ETP T-ALL patients with high MRD levels in the upcoming international pediatric trial (ALLTogether). Additionally, the future protocol for relapsed T-ALL in pediatric patients (IntreALL 2020) plans to incorporate venetoclax in combination with the current approach (UKALL-R3 based protocol).

4. The authors use glucose uptake as a parameter for glucose utilization. Although glucose uptake sometimes may correlate with glucose utilization, they are two different concepts. Metabolomics studies are required to track glucose flow to assess glucose utilization. And there are not such studies in the current manuscript.

We thank the reviewer for this point. We have performed glucose tracing over time and the results show a rapid incorporation of glucose-derived carbons into glycolytic intermediates in PI3K-driven leukemia cells, at a faster pace than wild-type cells. These new data are now included in the new Supplementary Figure 3.

5. Similarly, one cannot conclude that glutaminolysis is activated/inhibited from a metabolic screen showing decreased/elevated glutamine/glutamate levels. Assessing the glutamine flow is required to get such conclusion. And again, there are not such studies in the current manuscript.

We thank the reviewer for this highly relevant point. We have performed glutamine tracing over time in three PI3K-altered and three wild-type PDX *ex vivo* upon basal culture conditions or after glucose limitation. These new data support an active glutaminolytic flux in PI3K-altered leukemia over wild-type blasts, which is actively engaged upon glucose limitation to support TCA. These data are now part of the new figure 4 and supplementary figure 4. Altogether, we strongly believe that these new data are in line with our main message and that it strengthens the role of glutamine metabolism as the main salvage pathway of PI3K-driven leukemia to cope with glucose limitation.

Minor concerns:

There are not enough experimental details in the Figure legends/methods.

We have now completed the methods and the figure legends accordingly.